# SFS: Smarter Code Space Search improves LLM Inference Scaling

**Jonathan Light***[1],    **Yue Wu**[2], **Yiyou Sun**[3], **Wenchao Yu**[3], **Yanchi liu**[3], **Xujiang Zhao**[3],
**Ziniu Hu**[4] ,    **Haifeng Chen**[3],    **Wei Cheng**✉[3]
[1]Rensselaer Polytechnic Institute, [2]Princeton University, [3]NEC Laboratories America, [4]XAi
`lij54rpi.edu`

## ABSTRACT

We frame code generation as a black-box optimization problem within the code space and demonstrate how optimization-inspired techniques can enhance inference scaling. Based on this perspective, we propose SCATTERED FOREST SEARCH (SFS), a novel approach that improves solution diversity and better exploits feedback during evolutionary search. Our theoretical analysis illustrates how these methods help avoid local optima during optimization, leading to more efficient exploration. Extensive experiments on HumanEval, MBPP, APPS, CodeContests, and Leetcode reveal significant performance gains. For instance, our method achieves a pass@1 rate of 67.1% on HumanEval+ and 87.2% on HumanEval with GPT-3.5, marking improvements of 8.6% and 4.3% over the state-of-the-art, while also halving the iterations needed to find the correct solution. Furthermore, our approach scales more efficiently than existing search techniques, including tree search, line search, and repeated sampling.

## 1 INTRODUCTION

Recent work highlights the effectiveness of scaling inference compute over training compute (Snell et al., 2024; Brown et al., 2024; Gandhi et al., 2024). The most common approach by far is to repeatedly sample from the LLM with the same prompt and filter out the best response using a verifier, also known as best-of-N sampling (Cobbe et al., 2021; Lightman et al., 2023). Methods that leverage feedback from the verifier to revise previous solutions in a line or tree-like fashion have also been explored (Feng et al., 2023; Chen et al., 2024a). Code generation is one such setting where scaling LLM inference through repeated sampling has also been effective (Wang et al., 2024; Chen et al., 2022).

Inference scaling is effective because, given enough attempts, the LLM is likely to sample the correct solution eventually (Snell et al., 2024; Brown et al., 2024). Therefore, generating diverse solutions is crucial for effective exploration. Our experiments show that existing methods such as best-of-N (BoN) and tree search often produce similar solutions, leading to *insufficient exploration of the solution space* (refer to Sec.3.5). Hence, a sampling and testing approach that *balances exploration and exploitation* can greatly improve inference scaling.

Figure 1: **2D Visualization of Code Space** represents each point as a possible code solution. The goal is to efficiently search this space for the solution with the best performance, defined by the number of unit tests passed, as indicated by the contours above.

To tackle this issue, we propose framing solution generation as a **black-box optimization problem** (Golovin et al., 2017) (as illustrated in Figure 1), in which validation tests serve as the black box and the LLM functions as the optimizer (Yang et al., 2024). Drawing from optimization theory, we develop SCATTERED FOREST SEARCH (SFS) to efficiently search for code solutions that successfully pass the maximum number of validation tests through generating and evolving (refining) solutions.

---

*Work done during the internship at NEC Laboratories America. ✉Corresponding author.

Our method: 1) enhances exploration by enabling the LLM to propose diverse search directions, 2) improves exploitation by leveraging feedback and prior search experiences, and 3) initializes various random seed code solutions to ensure broader search coverage.

Specifically, SFS contains three key techniques. SCATTERING is a novel technique that dynamically varies input prompts when sampling from LLMs, driving more diverse and exploratory outputs. In SCATTERING, the LLM suggests different *textual optimization directions and steps*, analogous to *gradients* in numerical optimization, before advancing towards a new solution. During tree search refinement, SCATTERING effectively *perturbs or mutates* previous solutions, resulting in an **evolutionary search** process. We further propose FORESTING, the tree search equivalent of *multi-start optimization*, where SCATTERING plays a crucial role in *diversifying initialization seeds*, ensuring they are *well-distributed throughout the search space*. This enhances the breadth of exploration while effectively mitigating clearly incorrect solutions, such as those with syntax errors.

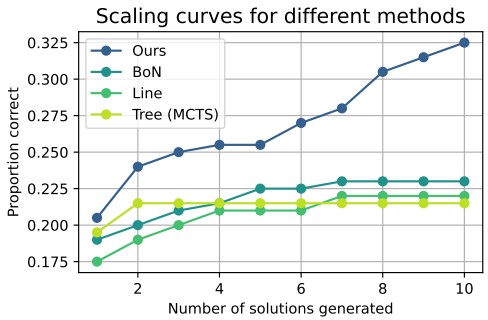

Figure 2: **Scaling curve for different search methods.** We run each method for 10 iterations total using `gpt-3.5-turbo` on APPS and report the proportion of problems where the correct solution has been discovered at each iteration.

Additionally, drawing inspiration from ant colony optimization and particle swarm optimization, we introduce SCOUTING to enhance SCATTERING by sharing feedback and experiences *across search branches*. When one branch discovers positive (or negative) results by following a specific textual direction, this information is relayed to guide future search steps, encouraging or discouraging exploration in that direction. Consequently, SCOUTING improves exploitation by *intensifying the search around promising textual directions*. We also provide a theoretical explanation demonstrating how our methods enhance exploration and circumvent local search regions when sampling from LLMs.

Our **parameter-free** method is simple yet effective. It does not require additional training or labelled data, yet achieves great performance. As illustrated in Figure 2, our method is able to discover the correct solution significantly faster than the other approaches. We evaluate our method on five different code generation benchmarks, HumanEval (Chen et al., 2021), MBPP (Austin et al., 2021), APPS, CodeContests, and Leetcode, showing increased inference time performance across the board, as well as better inference scaling. Our method also outperforms prior state-of-the-art techniques on HumanEval+ and MBPP+. Additionally, it generates higher *solution diversity* than previous methods, hence balancing exploration and exploitation, resulting in *faster discovery* of correct solutions and better scaling.

To sum up, our contributions are as follows:

- We frame code generation as a black-box optimization problem over the code/text space and demonstrate how gradient-free optimization techniques can be adapted to enhance inference scaling. This perspective highlights the importance of balancing exploration and exploitation in search-based code generation and the need for diverse generation.

- We introduce SCATTERED FOREST SEARCH (SFS), which applies optimization-inspired search techniques to improve exploration and avoid local optima. Specifically, SCATTERING and FORESTING enhance search diversity through textual optimization and seed initialization, while SCOUTING refines search efficiency by leveraging shared insights across search trajectories.

- We empirically validate our approach across multiple code generation benchmarks, including HumanEval, MBPP, Leetcode, APPS, and CodeContests, demonstrating significant improvements in accuracy, scalability, and solution diversity over existing search methods.

## 2 BACKGROUND

### 2.1 PROBLEM DESCRIPTION

In a program synthesis task $x = \langle p, H \rangle$, the solver is given a prompt $p$ in natural language, which asks the solver to write code for some object $s$. The goal is to complete the code implementation of $s$

Figure 3: **Overview of prior methods** used for code generation with LLMs. Points represent solutions. Hexagons represent initial solutions. Star represents the final selected solution.

| Best of N | Line search | Tree search |
|---|---|---|
| 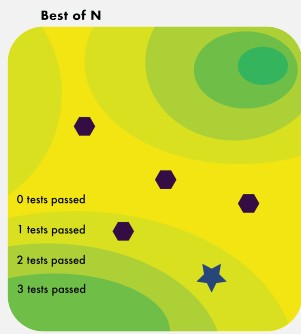 | 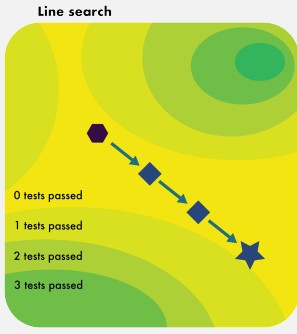 | 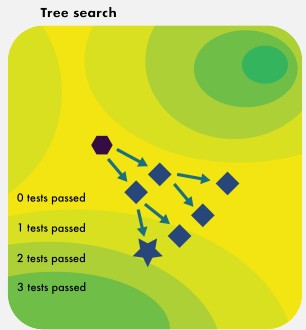 |
| Repeated sampling generates multiple solutions using the LLM without leveraging feedback from previous iterations. | Line search rigidly exploits feedback and cannot revert to a previous solution if a new change worsens the outcome | Tree search is more flexible but still lacks sufficient exploration, as the generated solutions tend to be very similar. |

such that it passes all the hidden tests $H$. The solver is not allowed to see the hidden tests. Sometimes, the solver is also given validation (visible) tests $V$ that they can test their solution $s$ on before they submit it for evaluation on the hidden tests $H$. They can also generate their own validation tests $V$. Usually both hidden and validation tests are in the form of a set of assert statements that test the functionalities of $s$. A description of one such code generation task is shown in Appendix B.

A solution $s'$ is said to be correct if it passes all the hidden tests $H$. The solver is allowed $k$ submissions $[s]_k$ to the hidden tests. If at least one submission $s^*$ passes all the hidden tests, then the task is considered to be solved. Given a set of tasks $\mathcal{X}$, the proportion of tasks $\langle p, H \rangle \in \mathcal{X}$ that are solved by the agent is called the **pass@k rate** (Chen et al., 2021).

## 2.2 PRIOR METHODS

A couple of different inference time methods have been tried in prior works to enhance the code generation capabilities of the LLM, which are shown in Figure 3. We elaborate more on the pros and cons of each method under a code space optimization framework below:

**Best of N** (BoN), or repeated sampling, involves sampling multiple independent solutions $[s]_n, s \sim$ LLM($p$) from a language model using the same prompt $p$. The best solution $s^*$ is then selected based on a verifier, commonly the number of validation tests passed (Li et al., 2022; Chen et al., 2024a).

**Line search** begins by sampling an initial seed code $s_0 \sim$ LLM($p$). It then iteratively refines the previous solution $s_{i-1}$ based on its test feedback $f_{i-1}$ (Shinn et al., 2023; Madaan et al., 2023). This iterative self-refinement leverages test execution feedback to guide the model toward sampling successful solutions, i.e. $s_i \sim$ LLM($p|s_{i-1}, f_{i-1}$). However, line search is limited by its rigidity – requiring improvements on the most recent solution, even if the latest edits are incorrect. Thus, it struggles to effectively explore the search space and is more likely to get stuck in local optima.

**Tree search** overcomes the rigidity of line search by generating multiple child solutions for each parent solution, utilizing tree-structured exploration methods such as BFS, DFS, and MCTS (Feng et al., 2023; Chen et al., 2024a; Hao et al., 2023; Yao et al., 2023; Zhou et al., 2024; Tian et al., 2024). Given a parent solution $s_i$ and its feedback $f_i$, the LLM produces $k$ child solutions $s_{i0}, s_{i1}, ..., s_{ik}$. Although higher temperatures can produce diverse solutions, in practice, these solutions often resemble each other because they originate from the same prompt (refer to Sec. 3.5). Consequently, tree search still faces challenges in fully exploring the search space.

## 3 METHODOLOGY

Our method incorporates three optimization inspired techniques to enhance both **exploration** and **exploitation** of tree search (MCTS) using LLMs. More details in Appendix C with pseudocode.

Figure 7: **Core techniques** used by SFS. Points represent solutions. Hexagons represent initial solutions. Star represents the final selected solution.

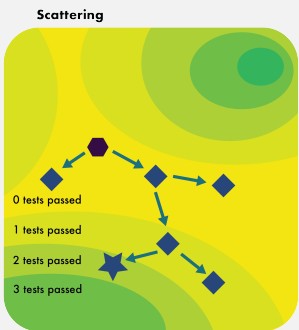
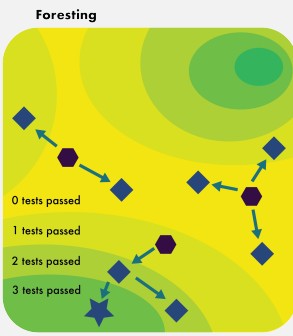
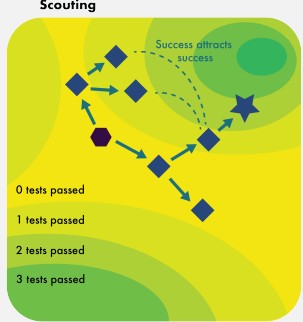

SCATTERING encourages tree search to **explore** more diverse solutions by using varied directional prompts for each branch or seed solution

FORESTING boosts **exploration** by performing tree search dynamically from multiple random seed solution starting points

SCOUTING shares successful search directions across branches of the search tree, providing general insights to better **exploit** feedback

## 3.1 TREE BRANCH SCATTERING

In tree search, children solutions of the same parent solution tend to be highly similar to one another since the LLM is given the same prompt to generate the children. We encourage more exploration when generating children solutions by querying the LLM to generate possible improvement directions $[d]_n$ first. The LLM is then instructed to implement a specific direction $d_j$ for each child $s_{ij}$ that it generates from parent $s_i$. This helps us explore different, often orthogonal, improvement directions that help us **explore a wider region** of the search space, similar to trust-region methods in numerical optimization (xiang Yuan, 2015). We refer to this technique as SCATTERING the tree branches.

$$\text{feedback from validation tests} \quad \rightarrow \quad \text{propose new textual directions using LLM} \quad \rightarrow \quad \text{choose 1 direction to implement when branching}$$

---

**Example SCATTERING Directions**

**Thoughts:** The feedback suggests that the main problem is that the function is returning the first element of `min-k` and `max-k` instead of the entire lists.

**Direction 1:** Modify the return statement to return `min-k` and `max-k` instead of `(min-k[0], max-k[0], min-k[-1], max-k[-1])`. This will ensure that the function returns the entire lists of minimum and maximum k elements.

**Direction 2:** Update the function to handle the case when K is greater than the length of the tuple. In this case, return the entire sorted tuple as both the minimum and maximum k elements.

---

Furthermore, we employ MCTS and utilize the UCT formula (Eq. 1) to dynamically **select directions for exploration**, as illustrated below

$$UCT(\boldsymbol{s}, \boldsymbol{d}) = \widehat{Q}(\boldsymbol{s}, \boldsymbol{d}) + c\sqrt{\frac{\ln\left(\sum_{\boldsymbol{b}} n(\boldsymbol{s}, \boldsymbol{b})\right)}{n(\boldsymbol{s}, \boldsymbol{d})}}, \tag{1}$$

where $c$ is the exploration parameter, $n(\boldsymbol{s}, \boldsymbol{d})$ is the number of visits of direction $\boldsymbol{d}$ at solution $\boldsymbol{s}$, and $\widehat{Q}(\boldsymbol{s}, \boldsymbol{d})$ is the estimated q-value which is updated via backpropagation as follows:

$$\widehat{Q}(\boldsymbol{s}_i, \boldsymbol{d}_{i+1})^{(t+1)} \leftarrow (1 - \alpha_n \widehat{Q}(\boldsymbol{s}_i, \boldsymbol{d}_{i+1})^{(t)} + \alpha_n \max\{\widehat{Q}(\boldsymbol{s}_i, \boldsymbol{d}_{i+1})^{(t)}, \widehat{Q}(\boldsymbol{s}_{i+1}, \boldsymbol{d}_{i+2})^{(t+1)}\} \tag{2}$$

where $\alpha_n$ is the weighted average parameter that depends on $n(\boldsymbol{s}, \boldsymbol{d})$. The backpropagation occurs along the entire MCTS simulated trajectory $\tau^{(t+1)} = [\boldsymbol{s}_0, \boldsymbol{d}_1, \boldsymbol{s}_2, ..., \boldsymbol{s}_{-2}, \boldsymbol{d}_{-1}, \boldsymbol{s}_{-1}]$, where the q-value for the penultimate state $\boldsymbol{s}_{-1}$ is updated using the value of the final state (% of validation tests passed) $\widehat{Q}(\boldsymbol{s}_{-2}, \boldsymbol{d}_{-1})^{(t+1)} \leftarrow v(\boldsymbol{s}_{-1})$. We take the max of $\widehat{Q}(\boldsymbol{s}_i, \boldsymbol{d}_{i+1})^{(t)}$ and $\widehat{Q}(\boldsymbol{s}_{i+1}, \boldsymbol{d}_{i+2})^{(t+1)}$ to ensure that if the next solution is worse, the current solution can be used instead. This approach dynamically selects which direction to explore, prioritizing more promising parent solutions $\boldsymbol{s}_j$ over exploring all directions of a parent solution $\boldsymbol{s}_i$. Using UCT to select distinct actions has been effective in balancing exploration and exploitation in other settings (Browne et al., 2012).

### 3.2 FOREST SEARCH AND FOREST SCATTERING

Iterative refinement faces the challenge that a very faulty initial seed solution may be difficult to correct effectively during the search process. An intuitive approach to address this issue is to generate multiple seed solutions and perform tree search from each one. We refer to this technique as FORESTING, in which we generate $n$ seed solutions $[s]_n$ and dynamically select which seed function to evolve from using the UCT formula (Eq. 1). Specifically, for each MCTS simulation, we select seed function $s_i$ with the highest UCT value and conduct the simulation from that point. This closely resembles **random seed initialization**, a widely and effective approach in optimization literature.

Similar to branch SCATTERING, we can also promote more diverse seed solution generation through forest SCATTERING by providing the LLM with different instructions prior to generating solutions. We investigate the impact of various SCATTERING instructions on performance in Sec. 3.5.

---

**Example Forest Seed Instructions**

**Seed instruction 1:** Write the code in a modular and extensible manner, ensuring each function or class has a single responsibility and can be easily extended or modified without impacting other parts of the system. Prioritize clear interfaces and loose coupling between components.

**Seed instruction 2:** Focus on writing highly efficient code with minimal memory usage and fast execution times. Use data structures and algorithms optimized for performance, and consider edge cases that could lead to bottlenecks. Prioritize speed and resource efficiency over readability.

**Seed instruction 3:** Prioritize readability and maintainability in your code. Write clear and descriptive comments, use meaningful variable and function names, and structure the code in a way that is easy for others to understand and modify. Follow established coding standards and best practices.

---

### 3.3 BRANCH SCOUTING

Inspired by optimization techniques like **Ant Colony Optimization** and **Particle Swarm Optimization**, we improve solutions by leveraging insights from effective improvement directions. After generating a new solution $s_{-1}$ using improvement directions $d_{-1}$ on $s_{-2}$, we provide feedback $f_{-1}$ to the LLM to assess its effectiveness and derive general insights. These insights are stored in global memory and included in future prompts, enabling shared knowledge of effective improvements across branches. This enhances feedback exploitation and strengthens our SCATTERING technique.

---

**Example SCOUTING Insights**

**Insight 1:** Modify the return statement to return `min-k` and `max-k` instead of (`min-k[0]`, `max-k[0]`, `min-k[-1]`, `max-k[-1]`). This will ensure that the function returns the entire lists of minimum and maximum k elements.

**Insight 2:** Update the function to handle the case when K is greater than the length of the tuple. In this case, return the entire sorted tuple as both the minimum and maximum k elements.

---

| use insights to gener-ate directions | $\rightarrow$ | see if the direction worked or not | $\rightarrow$ | update insights based on feedback |

### 3.4 A THEORETICAL PERSPECTIVE

The proposed techniques—SCATTERING and FORESTING —can be analyzed via the Markov chain theory, particularly focusing on the concepts of diverse transition kernels, conductance, and mixing times (Levin & Peres, 2020). We can define the search strategy as a Markov transition kernel $P(s, s')$ which denotes the probability of generating a new solution $s'$ given the current solution $s$. A chain of self-refined solutions $s_0, s_1, s_2, \ldots$ are generated following the transition kernel $P$.

In previous methods including line search and tree search, the transition is realized by an LLM $\pi$ that is conditioned on the previous solution $s$ and the feedback $f = F(s)$, denoted by $\pi(s'|s, f)$. The transition kernel is

$$P_{\text{previous}}(s, s') = \pi_c(s'|s, F(s)), \tag{3}$$

where we use $\pi_c$ to emphasize that the LLM $\pi$ is prompted to output code. We know $\pi(c)$ can be extremely concentrated and outputs highly similar solutions.

With SCATTERING, we first sample an improvement direction $d$ generated by the LLM with prompt $\pi(\cdot|s, F(s))$. And then prompt LLM $\pi$ to generate the next solution $s'$ given the current solution $s$,

the feedback $\boldsymbol{f} = F(\boldsymbol{s})$, and the improvement direction $\boldsymbol{d}$. The transition kernel is

$$P_{\text{SCATTERING}}(\boldsymbol{s}, \boldsymbol{s}') = \sum_{\boldsymbol{d}} \pi_t(\boldsymbol{d}|\boldsymbol{s}, F(\boldsymbol{s}))\pi_c(\boldsymbol{s}'|\boldsymbol{s}, F(\boldsymbol{s}), \boldsymbol{d}), \qquad (4)$$

where $\pi_c$ is still an extremely concentrated policy and outputs highly similar solutions. However, we have a text-based "reflection" $\pi_t(\boldsymbol{d}|\boldsymbol{s}, F(\boldsymbol{s}))$ that can generate highly diverse improvement directions. In practice, we can observe that indeed the directions are highly diverse.

In Markov chain theory, a diverse transition kernel increases the probability of moving between different regions of the state space (denoted by $\mathcal{S}$), enhancing the **conductance** $\Phi$ of the chain:

$$\Phi_P(S) = \frac{\sum_{\boldsymbol{s} \in S, \boldsymbol{s}' \notin S} \mu(\boldsymbol{s})P(\boldsymbol{s}, \boldsymbol{s}')}{\mu(S)}, \qquad (5)$$

where $S$ is a subset and $\mu$ is the stationary distribution. More details in Appendix M.

Higher conductance improves the spectral gap $\gamma$, which is inversely related to the mixing time of the Markov chain (see Cheeger's inequality) (Levin & Peres, 2020), reducing the likelihood of the chain getting trapped in local regions. More formally speaking, for a local region $S$, the previous methods (equation 3) rely on directly generating new responses that can easily stuck in the local region $S$ ($P_{\text{previous}}(\boldsymbol{s}, \boldsymbol{s}') \approx 0$ when $\boldsymbol{s} \in S$ and $\boldsymbol{s}' \notin S$, thus the conductance $\Phi$ is near 0). While our SCATTERING search (equation 4) can generate highly diverse directions $\boldsymbol{d}$ and lead to new solutions $\boldsymbol{s}'$ out of the local region $S$ ($P_{\text{SCATTERING}}(\boldsymbol{s}, \boldsymbol{s}') > 0$ if some $\boldsymbol{d}$ gives correct direction).

## 3.5 EMPIRICAL VALIDATION

We varied the types of seed instructions used to generate seed code during BoN sampling to validate the effects of increasing solution diversity with SCATTERING. In the **Jabberwocky** setting, the model was prompted with different lines from the humorous nonsense poem "Jabberwocky" before generating seed code. In the **Style** setting the model was prompted with different coding style instructions such as writing code in a 'highly modular way' or 'focus on brevity and clarity'. In the **Role** setting, the model was prompted with different software engineer personalities such as 'You are an innovator.' or 'You are a perfectionist'. All input prompts were LLM generated by `gpt-3.5-turbo` around a common theme.

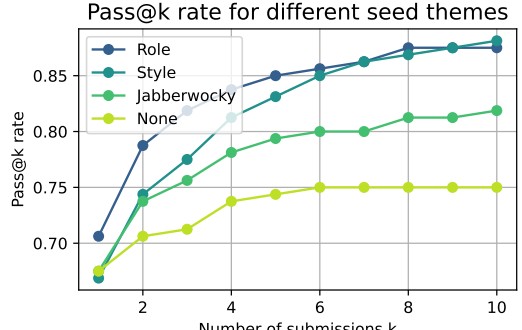

Figure 11: **Pass@k rate** for repeated sampling with different initialization seed types on HumanEval using `gpt-3.5-turbo-0613`. Increasing seed variety with SCATTERING significantly improves both Pass@k rate and scaling.

We present the pass@k performance of these seed generation styles in Figure 11, with detailed performance metrics in Tables 1 and 19. This includes: 1) the **pass@any** rate, which measures the proportion of problems where the correct solution was found at any point during the search, 2) the mean **validation score**, averaged across all candidate solutions generated (Eq. 12), and 3) the mean **BERT cosine similarity** between embeddings of candidate solution pairs, averaged over all problems. Embeddings were taken using CodeBERT, a pretrained model for understanding code semantically (Feng et al., 2020).

$$\text{Mean BERT cosine similarity} = \frac{1}{|\mathcal{X}|} \sum_{\langle \boldsymbol{p}, \boldsymbol{H}\rangle \in \mathcal{X}} \frac{1}{|\mathcal{S}_{\boldsymbol{p}}|(|\mathcal{S}_{\boldsymbol{p}}| - 1)} \sum_{\substack{\boldsymbol{s}, \boldsymbol{s}' \in \mathcal{S}_{\boldsymbol{p}} \\ \boldsymbol{s} \neq \boldsymbol{s}'}} \frac{\text{embed}(\boldsymbol{s}) \cdot \text{embed}(\boldsymbol{s}')}{\|\text{embed}(\boldsymbol{s})\|\|\text{embed}(\boldsymbol{s}')\|}$$

In addition to the metrics we discussed previously, we also include other similarity metrics such as **tf-idf similarity**, which measures the average cosine similarity between tf-idf vectors, the **Levenshtein similarity**, and the **token sequence similarity**. See App. J for more details, including examples.

As shown in Table 1, "Role" and "Style" performed best in discovering the correct solution. Surprisingly, even *unrelated prompts, like lines from nonsense poems, boosted performance*. Overall, all SCATTERING themes reduced solution similarity while maintaining comparable validation scores, demonstrating the effectiveness of using varied inputs during the search process.

Table 1: **Effects of different seed instruction SCATTERING themes.** 10 seed solutions were generated using `gpt-3.5-turbo` for each theme, and we filtered the seeds using 6 generated validation tests to select the best one to submit for evaluation on the HumanEval benchmark.

| Seed theme | pass@1 | pass@any | tf-idf sim. | BERT sim. | lev. sim. | seq. sim. | val. score |
|---|---|---|---|---|---|---|---|
| None | 72.5 | 75.0 | 0.9013 | 0.9976 | 0.8971 | 0.9361 | 0.7786 |
| Jabberwocky | 74.4 | 81.9 | 0.7559 | 0.9944 | 0.7749 | 0.8444 | 0.7658 |
| Style | 79.4 | 88.1 | 0.6826 | 0.9929 | 0.7119 | 0.7504 | 0.7548 |
| Role | 81.9 | 87.5 | 0.7734 | 0.9957 | 0.7907 | 0.8323 | 0.7649 |

## 4 EXPERIMENTS

We demonstrate that our search method outperforms prior methods by showing that it 1) achieves higher accuracy, 2) finds correct solutions faster and scales better, and 3) explores more diverse solutions without sacrificing exploitation of good ones.

### 4.1 EVALUATION BENCHMARKS

We evaluate our method on several popular code generation benchmarks. **HumanEval** consists of 164 human-generated Python questions (Chen et al., 2021), and **MBPP** includes 399 problems (Austin et al., 2021). The original sets included only 3 to 8 hidden tests $\mathcal{H}$, which researchers found inadequate for thoroughly evaluating correctness in edge cases. Additional hidden tests were added, resulting in the **HumanEval+** and **MBPP+** sets (Liu et al., 2024a).

Both **APPS** (Hendrycks et al., 2021) and **CodeContests** (Li et al., 2022) feature challenging code competition problems. From the 10,000 problems in APPS, we randomly sample 200 for evaluation due to budget constraints. We adapt the competition format of both datasets to resemble the HumanEval format for Python evaluation. **Leetcode** (Guo et al., 2024) includes recent problems scraped from the website, ensuring that LLMs released before July 2023 have not been trained on these data.

### 4.2 ACCURACY

We conduct experiments on a variety of code generation benchmarks as shown in Table 2, where we see that our method achieves a higher pass@1 rate than other search methods and the base accuracy (i.e., evaluating the first solution that the LLM model generates). Methods were given the same search budget (10 solutions), they used 6 self-generated validation tests.

Table 2: **Performance of our method compared to prior search methods.** Pass@1 performance reported here. Both solutions and validation tests were generated using `gpt-3.5-turbo`.

| Method / Benchmark | HumanEval+ | MBPP+ | Leetcode | APPS | CodeContests |
|---|---|---|---|---|---|
| Base | 58.5% | 64.9% | 30.0% | 16.0% | 1.82% |
| Line | 53.0% | 61.2% | 28.9% | 14.5% | 1.21% |
| Tree (MCTS) | 59.8% | 65.4% | 31.7% | 18.0% | 2.42% |
| Best of N | 65.2% | 64.4% | 33.3% | 19.5% | 1.82% |
| Ours (SFS) | **67.1%** | **65.7%** | **36.7%** | **20.5%** | **4.24%** |

We also measure the proportion of problems where the **correct solution was found (pass@any)** at any point during the search process, as shown in Table 3. We evaluate on HumanEval and MBPP rather than their plus versions due to computational constraints. HumanEval+ and MBPP+ contain 80x and 35x more tests, respectively, which makes it challenging to verify each generated solution. The pass@any rate is higher than pass@1 because, even if the correct solution is found, inaccurate validation tests may lead the algorithm to submit an alternative solution. Our method achieves significantly higher pass@any rates and demonstrates greater improvements from pass@1 to pass@any, indicating that it *can better leverage a more accurate verifier to filter for the correct solution*. We further explore the impact of noisy validation tests in detail in Sec. 4.6.

Table 3: **Pass@any accuracy of our method compared to prior search methods.** Both solutions and validation tests were generated using `gpt-3.5-turbo`. We run each method for 10 iterations.

| Method / Benchmark | HumanEval | MBPP | Leetcode | APPS | CodeContests |
|---|---|---|---|---|---|
| Line | 83.1% | 82.9% | 33.3% | 22.0% | 2.99% |
| Tree (MCTS) | 76.9% | 79.6% | 33.9% | 21.5% | 2.42% |
| Best of N | 75.6% | 77.3% | 33.9% | 23.0% | 3.03% |
| Ours (SFS) | **89.0%** | **86.1%** | **39.4%** | **32.5%** | **6.06%** |

We also compare to existing state-of-the-art works as shown in Table 4 and 5. In Table 4 setting, validation tests are self-generated, and cap our solution budget at 40 generations max, same as in prior literature (Zhou et al., 2024). In Table 5, a subset of the ground truth hidden tests are given (3 for HumanEval, 1 for MBPP) (Zhong et al., 2024), and we compare against similar methods under this setting. We see that our method achieves higher performance in both settings.

Table 4: **Comparison to prior works** when ground truth tests are not given. We report Pass@1 performance with GPT-3.5.

| Benchmark | HumanEval | MBPP |
|---|---|---|
| CoT (Wei et al., 2022) | 46.9 | 54.9 |
| ReAct (Yao et al., 2022) | 56.9 | 67.0 |
| Reflexion (Shinn et al., 2023) | 68.1 | 70.0 |
| ToT (Yao et al., 2023) | 54.4 | 65.8 |
| RAP (Hao et al., 2023) | 63.1 | 71.4 |
| LATS[a] (Zhou et al., 2024) | 75.6 | 79.6 |
| Ours (SFS) | **82.5** | **81.7** |

[a]ran under our setup, see App. E for similar setup

Table 5: **Comparison to prior works** when a subset of ground truth tests are given. We report Pass@1 performance with GPT-3.5.

| Benchmark Tests given | HumanEval 3 | MBPP 1 |
|---|---|---|
| SD (+Expl.) (Chen et al., 2023) | 81.1 | 74.4 |
| SD (+Trace) (Chen et al., 2023) | 80.5 | 72.6 |
| LDB (Zhong et al., 2024) | 82.9 | 76.0 |
| Ours (SFS) | **87.2** | **91.3** |

## 4.3 SCALABILITY

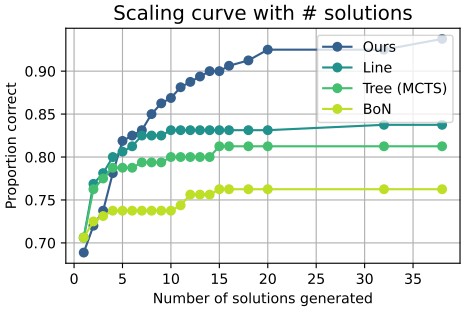
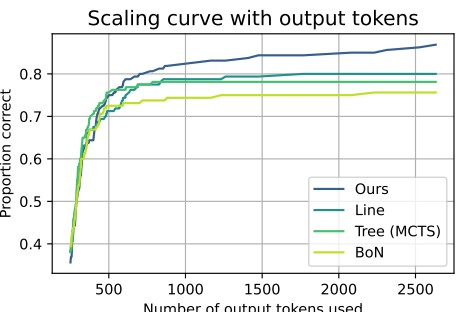

Figure 12: **Scaling curves for different search methods.** *Left:* Proportion of problems solved vs. number of solutions generated. *Right:* Proportion of problems solved vs. number of tokens used. Results are from `gpt-3.5-turbo` on HumanEval. Our method shows consistent improvement up to 20 solutions. Other methods plateau and do not catch up to SFS even with additional scaling. Additional curves can be found in Sec. D, including curves for other datasets.

We report the **average number of iterations** (solutions generated) it takes before the search algorithm discovers the correct solution in Table 6. **Iters. (incl)** is the average number of iterations it takes including problems where the algorithm succeeds on the first try. **Iters. (excl)** is the average number of iterations it takes excluding first try successes, where the search algorithm is actually used to find the correct solution. On both metrics, our method demonstrates the ability to *discover the correct solution much more quickly than the other methods*. Moreover, we see in Figure 2 and 12 that our method also *scales better than the other methods* on all datasets. Our method scales efficiently with higher budgets, improving up to 20 solutions while others plateau as shown in Figure 18.

Table 6: **Metrics for different search methods.** We run search methods for 10 iters. each using `gpt-3.5-turbo` on HumanEval. Our method generates more diverse solutions and discovers the correct solution faster. We also compare against a genetic algorithm (Romera-Paredes et al., 2024)

| Method | pass@1 | pass@any | BERT sim. | val. score | iters. (incl) | iters. (excl) |
|---|---|---|---|---|---|---|
| Line | 68.1% | 83.1% | 0.9992 | 0.795 | 2.09 | 7.13 |
| Tree (MCTS) | 75.6% | 76.9% | 0.9998 | 0.827 | 2.38 | 8.09 |
| Best of N | 73.8% | 75.6% | 0.9983 | 0.774 | 2.59 | 9.00 |
| Genetic | 74.4% | 75.6% | 0.9994 | 0.815 | 3.04 | 10.36 |
| Ours (SFS) | 82.5% | 89.0% | 0.9945 | 0.813 | 1.67 | 5.06 |

## 4.4 SOLUTION DIVERSITY

We see in Table 6 that our proposed search method is able to propose more diverse candidate solutions with a lower semantic similarity score, while maintaining high quality search with by generating solutions with high validation scores. This shows that our methods help both *exploration and exploitation without sacrificing too much of one for the other*. Detailed stats shown in App. G.

## 4.5 ABLATION ON TECHNIQUES

We performed an ablation study on the three introduced techniques as shown in Table 7 (and App. H), all of which enhanced performance and efficiency, with SCATTERING yielding the highest gains.

Table 7: **Ablation metrics for techniques used in our method.** We run search methods for 10 iterations each using `gpt-3.5-turbo` on HumanEval.

| Ablation | pass@1 | pass@any | BERT sim. | val. score | iters. (incl) | iters. (excl) |
|---|---|---|---|---|---|---|
| Everything | 82.5% | 89.0% | 0.9945 | 0.813 | 1.68 | 5.06 |
| NO SCATTERING | 75.6% | 78.1% | 0.9982 | 0.802 | 2.43 | 8.82 |
| NO FORESTING | 79.4% | 86.3% | 0.9982 | 0.817 | 2.05 | 6.56 |
| NO SCOUTING | 81.9% | 86.3% | 0.9942 | 0.792 | 2.12 | 6.05 |

Table 8: **Performance on benchmarks when the ground truth tests are given.** We run 10 iterations with our method using `gpt-3.5-turbo-0613` on HumanEval. When the tests are accurate, we can achieve even higher performance, even if only a subset is given.

| Tests given | pass@1 | pass@any | BERT sim. | val. score | iters. (incl) | iters. (excl) |
|---|---|---|---|---|---|---|
| No tests given | 82.5% | 89.0% | 0.9945 | 0.813 | 1.68 | 5.06 |
| 3 tests given | 87.2% | 90.2% | 0.9952 | 0.862 | 2.34 | 6.86 |
| All tests given | 89.0% | 90.2% | 0.9949 | 0.864 | 2.04 | 5.88 |

## 4.6 VERIFIER (VALIDATION TEST) ACCURACY

Previous work emphasized the importance of a reliable verifier (Liu et al., 2024a; Chen et al., 2022; Zhang et al., 2023a). In our case, we use self-generated validation tests noisy **as a black-box verifer**. Figure 13 shows the confusion matrix for whether self-generated validation tests accurately predict solution correctness. The false negative rate of 27.5% highlights misalignment between validation and ground truth tests. We performed an ablation study comparing performance when given different numbers of ground-truth tests as validation (Table 8). While finding the best verifier is not our focus, the results show a significant impact of verifier accuracy on pass@1. There are strong interactions between the search method and evaluation metric—just one ground-truth validation test dramatically improves performance in Table 5. However, the 33.75% inaccuracy of validation tests do not affect the pass@any rate much as shown in Table 8, which demonstrates the *robustness of our method to validation noise*. Detailed metrics shown in App. I.

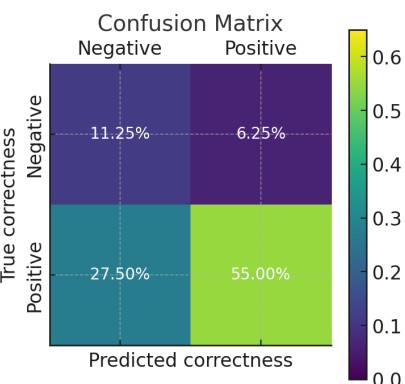

Figure 13: **Confusion matrix for self-generated validation tests** with `gpt-3.5-turbo-0613` using our method.

### 4.7 ABLATION ON MODEL

Fig. 14 shows weaker models scale better with our method, highlighting a trade-off: better-trained models scale worse with inference compute. This suggests inference-time scaling compensates for limited training quality, benefiting weaker models. Detailed stats are in App. F.

## 5 ADDITIONAL RELATED WORK

**Code generation with large language models.** Some works focus on either training their own model (Guo et al., 2024) or fine tuning existing models to adapt them towards code related tasks (Roziere et al., 2023; Jain et al., 2023; Roziere et al., 2023). Recent literature has shown a flora of methods to improve LLM code generation performance during inference time, including agentic approaches (Qian et al., 2023; Liu et al.,

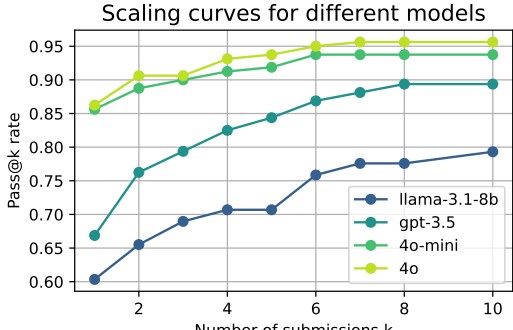

Figure 14: **Scaling curve for different LLM-models.** We run our method on each model for 10 iterations on HumanEval and report the proportion of problems where the correct solution has been discovered at each iteration.

2024b), multi-agent approaches (Tao et al., 2024; Hong et al., 2023; Islam et al., 2024), tool usage (Zhang et al., 2024), and self-repair (Le et al., 2023; Olausson et al., 2023a). There is a trade-off between scaling inference and training compute, with an optimal balance between them (Wu et al., 2024; Sardana et al., 2024). Our work complements this by demonstrating the impact of enhanced solution diversity and exploration on scaling. While this work focuses on MCTS, our method can seamlessly integrate with advanced solution sampling methods during refinement (Tang et al., 2024).

**Solution validation and feedback.** Prior work has demonstrated that effective and accurate validation methods significantly boost performance in code generation (Chen et al., 2022; 2024b) and related tasks like math theorem proving (Cobbe et al., 2021; Uesato et al., 2022). The importance of a reliable evaluation metric is critical (Liu et al., 2024a), and incorporating natural language feedback and reflection on execution results further enhances performance (Zhang et al., 2023c; Bai et al., 2022). Our work complements this by introducing new search methods that more effectively discover solutions meeting these validation and evaluation criteria. Combining a strong search method with a robust validation approach leads to improved solution generation performance.

**Black box optimization.** While many of these methods are domain specific, we adapt many of the insights behind these methods into using LLMs to search over language and code space. Our textual optimization steps can be thought of as 'textual gradients', though we do not use them to conduct backpropogation (Yuksekgonul et al., 2024). There have been some recent works exploring the usage of LLMs as optimizers for different problems (Zhang et al., 2023b; Liu et al., 2024c), including using evolutionary optimization algorithms (Lange et al., 2024; Liu et al., 2024b).

**Tree search.** Tree search has proven effective in decision-making domains. Previous works have applied tree search to multi-step tasks, such as determining the next reasoning step (Silver et al., 2017; Zhou et al., 2024; Yao et al., 2023; Besta et al., 2024) or generating the next code token (Zhang et al., 2023c). While prior work primarily uses tree search for planning (Jiang et al., 2023; Feng et al., 2023; Bairi et al., 2024), we show that *tree search can also be used for optimization*, functioning as a black-box method for exploring a region rather than a line.

## 6 CONCLUSION

We have shown that framing code generation as an optimization task over the code space and applying SCATTERED FOREST SEARCH is highly effective. The simple yet powerful SCATTERING technique, which encourages the LLM to produce more diverse outputs, underscores the importance of exploration in optimization and search processes. This framework and these techniques can offer valuable insights for other code or language generation tasks.

The integration of search-based methods could significantly reduce computational costs while enhancing performance, making them attractive for large-scale deployments, especially in real-time applications or those requiring resource efficiency. This framework also lays the groundwork for future research into optimization strategies for language models, potentially leading to more advanced search algorithms, hybrid models, and novel techniques that push the limits of generative models.

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

# APPENDIX: SMARTER CODE SPACE EXPLORATION WITH LLMS

## A    ETHICS STATEMENT

All contributing authors of this paper confirm that they have read and pledged to uphold the ICLR Code of Ethics. Our proposed method is specifically designed for code generation using LLMs. We acknowledge the potential for LLMs to generate code that may inadvertently introduce vulnerabilities or ethical concerns. Our research prioritizes the development of methodologies that emphasize responsible usage, ensuring that generated code adheres to best practices in security and ethics. We recognize that LLMs can perpetuate and amplify biases present in training data. We aim to contribute positively to the field of code generation while addressing the potential challenges and responsibilities that arise from the use of advanced AI technologies.

## B    PROBLEM SETTING AND CODING PROBLEM EXAMPLE

In the program synthesis task $x = \langle p, H \rangle$, the solver is provided with a problem prompt $p$, described in natural language, which requests the generation of code to implement a specific functionality or object $s$. The objective is to generate a solution $s'$ that passes all the hidden tests $H$ designed to evaluate its correctness. These hidden tests $H$ are not visible to the solver under any circumstances. In some cases, the solver may also have access to a set of validation tests $V$, which they can use to verify their solution $s'$ before submitting it for evaluation against the hidden tests. Alternatively, the solver may generate their own validation tests $V$ to test the functional correctness of $s'$. Both hidden and validation tests are typically structured as a set of assert statements validating specific functionalities of $s$. A representative example of this problem setup is provided below.

---

**Example code generation prompt, solution, and tests**

**Prompt:** Write a function `greatest_common_divisor(a,b)` that returns the GCD of two integers `a` and `b`

**Validation tests:**
```
assert(greatest_common_divisor(3,5) == 1)
assert(greatest_common_divisor(25,15) == 5)
assert(greatest_common_divisor(0,3) == 3)
```

**Proposed solution 1:**
```
def greatest_common_divisor(a, b):
    for i in range(min(a, b), 0, -1):
        if a % i == 0 and b % i == 0:
            return i
```
**Test feedback:**
```
assert(greatest_common_divisor(3,5) == 1) #output 1 is
correct
assert(greatest_common_divisor(25,15) == 5) #output 5 is
correct
assert(greatest_common_divisor(0,3) == 3) #output None is
incorrect
```

---

A solution $s'$ is deemed correct if and only if it passes all hidden tests $H$. Solvers are permitted up to $k$ attempts $[s]_k$ to submit their solutions for evaluation. If at least one submission $s^*$ satisfies all hidden tests, the task is considered solved. Given a set of tasks $\mathcal{X}$, the proportion of tasks $\langle p, H \rangle \in \mathcal{X}$ successfully solved by the agent within $k$ submissions is referred to as the **pass@k rate** (Chen et al., 2021).

### B.1 JUSTIFICATION FOR PROBLEM SETUP

In our problem setting, the solver is explicitly not allowed to access the hidden tests $\boldsymbol{H}$. This constraint ensures that the task closely mirrors real-world applications of program synthesis, where the solver must produce solutions based solely on the problem prompt $\boldsymbol{p}$. Such a setup aligns with platforms like LeetCode or conventional coding assessments, where test cases remain hidden to preserve the integrity of the evaluation process and discourage overfitting to known test sets.

This restriction is not only practical but also reflects the needs of end users in real-world scenarios. Writing comprehensive and correct unit tests is a time-consuming process, requiring substantial domain knowledge and effort. In practice, users would prefer tools that can autonomously generate correct implementations from a prompt without requiring additional test specifications or guidance. By requiring solvers to rely on self-generated validation tests or implicit reasoning to produce correct solutions, this setup encourages the development of tools capable of robust and generalized problem-solving.

By prioritizing correctness under these constraints, our framework evaluates an agent's ability to understand the problem and synthesize accurate solutions in a realistic and user-centric manner, making it an ideal benchmark for program synthesis tasks.

## C  METHODOLOGY DETAILS

### C.1  TRADITIONAL MONTE CARLO TREE SEARCH (MCTS)

Monte Carlo Tree Search (MCTS) is a widely used algorithm for decision-making in environments with large, complex state spaces, such as games and optimization problems. The algorithm incrementally builds a search tree by simulating possible outcomes and leveraging these simulations to balance **exploration** (visiting new states) and **exploitation** (refining knowledge about promising states). The MCTS process consists of four main stages: **selection**, **expansion**, **simulation**, and **backpropagation**.

#### C.1.1  SELECTION

The algorithm begins at the root node, corresponding to the current state $\boldsymbol{s}$, and iteratively selects child nodes using a selection policy designed to balance exploration and exploitation. The most commonly used selection policy is the Upper Confidence Bound for Trees (UCT), which evaluates each child state $\boldsymbol{s}'$ and corresponding action (or direction) $\boldsymbol{d}$ according to the formula:

$$UCT(\boldsymbol{s}, \boldsymbol{d}) = \widehat{Q}(\boldsymbol{s}, \boldsymbol{d}) + c\sqrt{\frac{\ln\left(\sum_{\boldsymbol{b}} n(\boldsymbol{s}, \boldsymbol{b})\right)}{n(\boldsymbol{s}, \boldsymbol{d})}}, \tag{6}$$

where $\widehat{Q}(\boldsymbol{s}, \boldsymbol{d})$ is the estimated value of taking action $\boldsymbol{d}$ from state $\boldsymbol{s}$, $n(\boldsymbol{s}, \boldsymbol{d})$ is the number of times this action has been visited, and $c$ is a hyperparameter controlling the exploration-exploitation trade-off. This equation encourages actions that either have high value (exploitation) or have been visited less frequently (exploration).

#### C.1.2  EXPANSION

After a leaf node is reached during selection, one or more child nodes representing potential future states $\boldsymbol{s}'$ are added to the search tree. These states are typically generated by sampling actions or improvement directions $\boldsymbol{d}$ from a predefined set or policy. The added nodes represent unexplored areas of the search space.

#### C.1.3  SIMULATION

Once a new node is expanded, the algorithm performs a simulation or rollout from that state $\boldsymbol{s}'$. In this step, a playout policy is used to sample a trajectory of states and actions until a terminal state or evaluation horizon is reached. The outcome of this simulation, denoted as $v(\boldsymbol{s}')$, provides an estimate of the quality of the state $\boldsymbol{s}'$. This value could represent a game outcome (e.g., win/loss), a performance metric (e.g., number of test cases passed), or another problem-specific objective.

### C.1.4 BACKPROPAGATION

The results of the simulation are then propagated back up the search tree to update the value estimates $\widehat{Q}(\boldsymbol{s}, \boldsymbol{d})$ for the nodes along the trajectory $\tau = [\boldsymbol{s}_0, \boldsymbol{d}_1, \boldsymbol{s}_2, \ldots, \boldsymbol{s}_{-1}]$. The value update is performed iteratively using a weighted average, which incorporates the new simulation result while retaining information from prior simulations:

$$\widehat{Q}(\boldsymbol{s}_i, \boldsymbol{d}_{i+1})^{(t+1)} \leftarrow (1 - \alpha_n)\widehat{Q}(\boldsymbol{s}_i, \boldsymbol{d}_{i+1})^{(t)} + \alpha_n \max\{\widehat{Q}(\boldsymbol{s}_i, \boldsymbol{d}_{i+1})^{(t)}, \widehat{Q}(\boldsymbol{s}_{i+1}, \boldsymbol{d}_{i+2})^{(t+1)}\}, \quad (7)$$

where $\alpha_n$ is a weighting parameter that depends on the visit count $n(\boldsymbol{s}, \boldsymbol{d})$, and $\max$ ensures that only the best-performing solutions along the trajectory are prioritized.

## C.2 SCATTERED FOREST SEARCH

---

**Algorithm 1:** Scattered Forest Search (SFS) with SCOUTING

---

**Input:** Language Model $\mathcal{L}$, Prompt $\boldsymbol{p}$, Number of iterations $N$, Branching factor $k$, Exploration parameter $c$
**Output:** Best solution $s^*$

**Function** `Scattering`($node$)**:**
  Generate diverse directions $\{d_1, d_2, \ldots, d_k\}$ using $\mathcal{L}$ conditioned on feedback for $node$;
  Save directions $\{d_1, d_2, \ldots, d_k\}$ to $node.directions$, with $node.qvalue(d_i) \leftarrow 0$;
  **return** $node.directions$;

**Function** `Foresting`($\mathcal{S}_0$)**:**
  Initialize forest with random seed solutions $\{s_1, s_2, \ldots, s_n\}$ sampled by $\mathcal{L}$;
  **foreach** $seed \in \mathcal{S}_0$ **do**
    │  `Scattering`($seed$); // Generate directions for each seed
  **end**
  **return** forest $\mathcal{S}_0$;

**Function** `Scouting`($parent, direction, child, feedback, \mathcal{I}$)**:**
  Update global insights $\mathcal{I}$ using feedback from applying $direction$ to $parent$, resulting in $child$;

**Function** `SelectSeed`($\mathcal{S}$)**:**
  Select the most promising seed $s$ from $\mathcal{S}$ using UCT formula (Eq. 1);
  **return** $s$;

**Function** `SelectDirection`($node$)**:**
  Select the most promising direction $d$ from $node.directions$ using UCT formula;
  **return** $d$;

**Function** `Simulate`($seed$)**:**
  $parent \leftarrow seed, \tau \leftarrow \emptyset$;
  **while** *parent is not a leaf node* **do**
    │  $direction \leftarrow$ `SelectDirection`($parent$);
    │  $child \leftarrow$ Child node corresponding to $direction$;
    │  $\tau.append((parent, direction, child))$;
    │  $parent \leftarrow child$;
  **end**
  **return** leaf node and trajectory $\tau$;

**Function** `Expand`($parent, direction$)**:**
  Generate new solution $\boldsymbol{s}'$ using $\mathcal{L}$ conditioned on $\boldsymbol{s}$, $direction$, and feedback;
  Create new child node $child$ with solution $\boldsymbol{s}'$ and feedback from validation tests on $parent$;
  **return** $child$;

**Function** `Backpropagate`($\tau, leaf, reward$)**:**
  $leaf.qvalue[direction] \leftarrow reward$;
  **foreach** $(parent, direction, child) \in \tau$ *(in reverse order)* **do**
    │  Update $parent.qvalue[direction]$ using Eq. 2;
    │  $parent.visits[direction] \leftarrow parent.visits[direction] + 1$;
  **end**

**begin**
  Initialize global insights $\mathcal{I} \leftarrow \emptyset$;
  Initialize forest $\mathcal{S}_0 \leftarrow$ `Foresting`();
  Generate validation tests $\boldsymbol{V}$ if not given;
  **for** $i \leftarrow 1$ **to** $N$ **do**
    │  $seed \leftarrow$ `SelectSeed`($\mathcal{S}_0$);
    │  $leaf, \tau \leftarrow$ `Simulate`($seed$);
    │  $direction \leftarrow$ `SelectDirection`($leaf$);
    │  $child \leftarrow$ `Expand`($leaf, direction$);
    │  $feedback, reward \leftarrow$ Execution feedback and validation score of $child$;
    │  `Backpropagate`($\tau, leaf, reward$);
    │  `Scouting`($leaf, direction, child, feedback, \mathcal{I}$);
    │  `Scattering`($child$); // Generate directions for the new node
  **end**
  $s^* \leftarrow$ Best solution across all trees;
**end**

---

**Python Pseudocode for SFS with SCOUTING**

```python
def scattering(node):
    """
    Generate diverse directions for a given node using the language model (LM).
    """
    directions = [d1, d2, ..., dk]  # Generate using LM conditioned on node feedback
    node.directions = directions
    node.qvalue = {d: 0 for d in directions}  # Initialize q-values for each direction
    return directions

def foresting():
    """
    Initialize a forest with multiple random seed solutions and generate directions for each seed.
    """
    forest = [s1, s2, ..., sn]  # Generate initial seeds using LM
    for seed in forest:
        scattering(seed)  # Generate directions for each seed
    return forest

def scouting(parent, direction, child, feedback, insights):
    """
    Update global insights using feedback from applying a direction to the parent to produce the child.
    """
    # Update insights with the feedback and interaction details
    insights.update({(parent, direction): feedback})

def select_seed(forest):
    """
    Select the most promising seed from the forest using the UCT formula.
    """
    return max(forest, key=lambda s: uct_value(s))  # Replace with actual UCT calculation

def select_direction(node):
    """
    Select the most promising direction from a node's directions using the UCT formula.
    """
    return max(node.directions, key=lambda d: uct_value(node, d))  # Replace with actual UCT calculation

def simulate(seed):
    """
    Simulate a path starting from a seed, selecting the best direction at each step.
    """
    parent = seed
    trajectory = []
    while not is_leaf_node(parent):
        direction = select_direction(parent)
        child = get_child_node(parent, direction)
        trajectory.append((parent, direction, child))
        parent = child
    return parent, trajectory

def expand(parent, direction):
    """
    Expand the search tree by applying a direction to the parent and generating a new solution.
    """
    new_solution = generate_solution(parent.solution, direction)  # Use LM
    feedback = validate_solution(new_solution)  # Validation feedback
    child = create_node(new_solution, feedback)  # Create a new child node
    return child

def backpropagate(trajectory, leaf, reward):
    """
    Backpropagate the reward along the trajectory.
    """
    leaf.qvalue[direction] = reward
    for parent, direction, child in reversed(trajectory):
        parent.qvalue[direction] = backprop_update(reward)
        reward = parent.qvalue[direction]
        parent.visits[direction] += 1  # Increment visit count

# Main algorithm
def scattered_forest_search(model, prompt, iterations, branching_factor, exploration_param):
    """
    Perform the Scattered Forest Search (SFS) algorithm.
    """
    insights = {}  # Initialize global insights
    forest = foresting()  # Initialize forest of seed solutions
    validation_tests = generate_validation_tests()  # Generate tests if not given

    for _ in range(iterations):
        seed = select_seed(forest)
        leaf, trajectory = simulate(seed)
        direction = select_direction(leaf)
        child = expand(leaf, direction)
        feedback, reward = execute(child, validation_tests)  # Run validation tests and get reward
        backpropagate(trajectory, leaf, reward)
        scouting(leaf, direction, child, feedback, insights)
        scattering(child)  # Generate directions for the new node

    best_solution = get_best_solution(forest)
    return best_solution
```

SFS integrates three key components: **scattering**, **foresting**, and **scouting**, with tree search performed using Monte Carlo Tree Search (MCTS). Below, we detail the main components and steps of the algorithm.

### C.3    ALGORITHM OVERVIEW

SFS operates over multiple iterations, progressively refining the solution space using a search tree initialized with diverse seed solutions. For each iteration, SFS balances exploration and exploitation using the Upper Confidence Bound for Trees (UCT) formula to guide the search. The algorithm begins by initializing global insights $\mathcal{I}$, a forest of seed solutions $\mathcal{S}_0$, and validation tests $\boldsymbol{V}$. It then performs the following steps over $N$ iterations.

**Initialization**

- **Global Insights**: A global memory $\mathcal{I}$ is initialized to store feedback and general insights gained during the search process, enabling knowledge sharing across branches.
- **Forest Creation**: The algorithm generates a set of initial seed solutions $\mathcal{S}_0 = \{\boldsymbol{s}_1, \boldsymbol{s}_2, \ldots, \boldsymbol{s}_n\}$ by sampling diverse solutions using the language model $\mathcal{L}$. For each seed:
    - **Direction Initialization**: Diverse directions $\{d_1, d_2, \ldots, d_k\}$ are generated for each seed solution by prompting $\mathcal{L}$ with feedback from the seed.
    - **Q-Values**: **Q-values for each direction are initialized to 0**, i.e., $node.qvalue(d_i) \leftarrow 0$. These values are updated dynamically based on feedback during backpropagation.

**Seed Selection**    The algorithm selects the most promising seed solution $\boldsymbol{s}$ from the forest $\mathcal{S}$ using the UCT formula:

$$UCT(\boldsymbol{s}, \boldsymbol{d}) = \widehat{Q}(\boldsymbol{s}, \boldsymbol{d}) + c\sqrt{\frac{\ln\left(\sum_{\boldsymbol{b}} n(\boldsymbol{s}, \boldsymbol{b})\right)}{n(\boldsymbol{s}, \boldsymbol{d})}}, \tag{8}$$

where $c$ is the exploration parameter, $n(\boldsymbol{s}, \boldsymbol{d})$ is the visit count for direction (or seed) $\boldsymbol{d}$ at state $\boldsymbol{s}$, and $\widehat{Q}(\boldsymbol{s}, \boldsymbol{d})$ is the estimated reward for that direction. This balances exploration of under-explored solutions and exploitation of high-reward solutions. *Unvisited seeds/directions always have infinite UCT values, which means that we will always explore unvisited seeds or directions first.*

**Simulation**    Starting from the selected seed $\boldsymbol{s}$, the algorithm simulates a trajectory $\tau = [(\boldsymbol{s}_i, \boldsymbol{d}_i, \boldsymbol{s}_{i+1})]$ by iteratively selecting the most promising direction $\boldsymbol{d}_i$ using the UCT formula. The simulation proceeds until a leaf node is reached.

**Expansion**    At the leaf node, the algorithm expands the search tree by generating a new child node:

$$\boldsymbol{s}' = \mathcal{L}(\boldsymbol{s}, \boldsymbol{d}, \text{feedback}), \tag{9}$$

where $\mathcal{L}$ generates a new solution $\boldsymbol{s}'$ by incorporating the parent solution $\boldsymbol{s}$, the improvement direction $\boldsymbol{d}$, and validation feedback. The newly generated solution $\boldsymbol{s}'$ is validated, and the proportion of validation tests passed serves as the **reward**:

$$\text{Reward} = \frac{\text{Number of tests passed}}{\text{Total number of tests}}. \tag{10}$$

**Backpropagation**    The reward is backpropagated along the trajectory $\tau$:

$$\widehat{Q}(\boldsymbol{s}_i, \boldsymbol{d}_{i+1})^{(t+1)} \leftarrow (1 - \alpha_n)\widehat{Q}(\boldsymbol{s}_i, \boldsymbol{d}_{i+1})^{(t)} + \alpha_n \max\{\widehat{Q}(\boldsymbol{s}_i, \boldsymbol{d}_{i+1})^{(t)}, \widehat{Q}(\boldsymbol{s}_{i+1}, \boldsymbol{d}_{i+2})^{(t+1)}\}, \tag{11}$$

where $\alpha_n$ is a weighting parameter dependent on visit counts. **The visit count $n(\boldsymbol{s}, \boldsymbol{d})$ for each direction is incremented by 1** during backpropagation.

**Scouting**    The feedback $\boldsymbol{f}$ from applying a direction $\boldsymbol{d}$ to a parent node $\boldsymbol{s}$ is analyzed to extract general insights. These insights are stored in global memory $\mathcal{I}$ and reused to guide future direction generation. Scouting improves the search's efficiency by leveraging knowledge gained from past iterations.

**Scattering**    For every expanded node, diverse new improvement directions $\{d_1, d_2, \ldots, d_k\}$ are generated using $\mathcal{L}$. This encourages exploration of orthogonal regions of the solution space, reducing the likelihood of stagnation in local optima.

**Final Selection**   After $N$ iterations, the best solution $s^*$ across all trees in the forest is selected. The best solution is determined based on the highest reward, i.e., the **proportion of validation tests passed**.

## C.4   KEY FEATURES OF SFS

- **Q-Value Initialization**: For every new direction, **Q-values are initialized to 0** and dynamically updated during backpropagation.
- **Reward Definition**: The reward is explicitly defined as the **proportion of validation tests passed**, providing a consistent measure of solution quality.
- **Diversity**: By incorporating scattering and foresting, the algorithm ensures diversity in both initial seed solutions and branching directions, reducing the risk of stagnation in local optima.
- **Global Insights**: The scouting mechanism enables the sharing of effective improvement strategies across the search space, enhancing the algorithm's exploitation capabilities.

The SFS algorithm's integration of scattering, foresting, and scouting represents a powerful approach to systematically navigate and optimize large solution spaces.

## C.5   ALGORITHM COMPLEXITY

The time complexity of the Scattered Forest Search (SFS) algorithm is primarily determined by the number of iterations $N$ and the traversal steps involved in each iteration.

**Time Complexity:**   At each iteration, the algorithm performs several steps: selecting seeds, simulating a trajectory, expanding nodes, and backpropagating rewards. In the worst-case scenario, the depth of the solution tree is proportional to $N$, resulting in $O(N)$ steps for traversal. Over $N$ iterations, this results in a total time complexity of $O(N^2)$. Additionally:

- **Scouting and Scattering:** The scouting and scattering steps at each iteration involve generating new directions and updating global insights, which have constant time complexity per iteration.
- **Comparison to Other Methods:**
  - *Line Search:* Line search explores solutions sequentially without tree traversal, yielding a time complexity of $O(N)$.
  - *Beam of Nodes (BoN):* Similar to line search, BoN has a time complexity of $O(N)$ due to its focus on maintaining and updating a fixed number of candidate solutions.
  - *Tree Search (MCTS):* Traditional Monte Carlo Tree Search also has a time complexity of $O(N^2)$, as it involves tree traversal similar to SFS.

Thus, the SFS algorithm shares the same $O(N^2)$ time complexity as tree search but differs significantly in its use of global insights and scattering to improve solution diversity and efficiency.

**Space Complexity:**

- *SFS:* The algorithm stores all generated solutions, resulting in a space complexity of $O(N)$. In addition, global insights $\mathcal{I}$ require constant space $O(1)$ since only the most recent updated insights need to be kept, and insights are kept within a fixed length.
- *Line Search:* Line search has a space complexity of $O(1)$, as it retains only the most recent solution.
- *BoN and Tree Search:* Both methods, like SFS, require $O(N)$ space to store all candidate solutions.

While the memory overhead for SFS is comparable to other tree-based methods, the efficient use of global insights enhances the algorithm's performance without additional memory costs.

**Practical Considerations:**   Although SFS involves tree traversal, which contributes to the $O(N^2)$ complexity, the computational bottleneck in practice is the solution generation process, typically

dominated by LLM API calls. Thus, the tree traversal cost has a negligible impact on runtime in practical settings.

## C.6 LLM REQUEST COST

Our algorithm requires **two LLM-API calls** (requests) per solution generated. One API call is used for code generation based on SCATTERING improvement directions. One API call is used for reflection after the code is generated and executed. The reflection step involves (1) updating the global insights based on the execution feedback (2) generating new SCATTERING directions based on the global insights and execution feedback. In practice, the two components of the reflection step can be separated into two API calls run in parallel to increase computational efficiency and code cleanness, which is how we implemented it.

# D  SCALING

Our method scales well with increased iterations. We show scaling curves for each dataset as shown in Figures 15, 16, 17. We also display scaling curves at higher budgets as shown in Figure 18. We see that performance mostly plateaus after the 20 solutions are generated. Moreover, other methods do not catch up to our method each with higher search budgets, suggesting how our method is able to fundamentally better explore the solution spaces, including solutions that other methods will not consider.

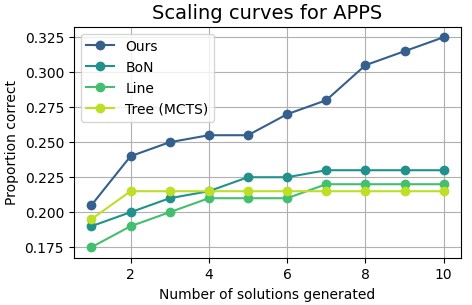 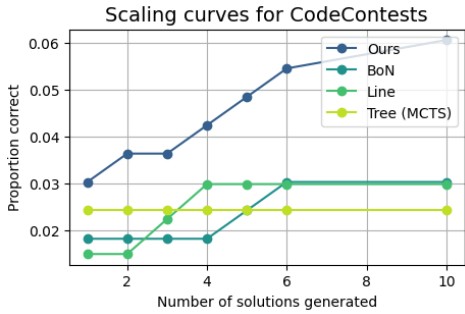

Figure 15: **Scaling curves for different search methods on APPS (left) and CodeContests (right).** We run each method for 10 iterations using `gpt-3.5-turbo`, reporting the proportion of problems where the correct solution is discovered at each iteration.

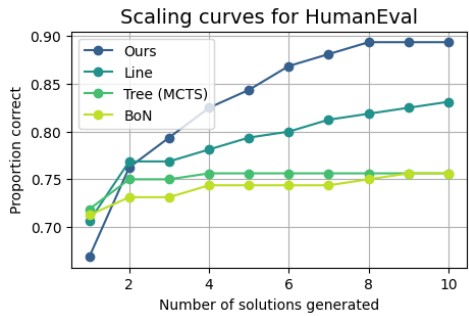 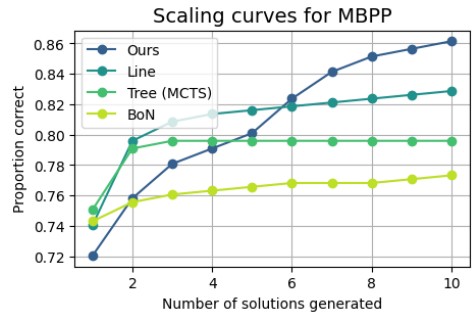

Figure 16: **Scaling curves for different search methods on HumanEval (left) and MBPP (right).** We run each method for 10 iterations using `gpt-3.5-turbo`, reporting the proportion of problems where the correct solution is discovered at each iteration.

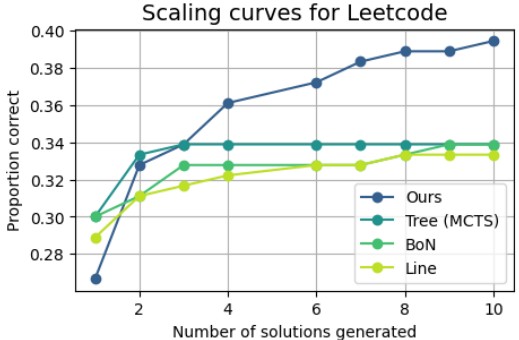

Figure 17: **Scaling curves for different search methods on LeetCode.** We run each method for 10 iterations using `gpt-3.5-turbo`, reporting the proportion of problems where the correct solution is discovered at each iteration.

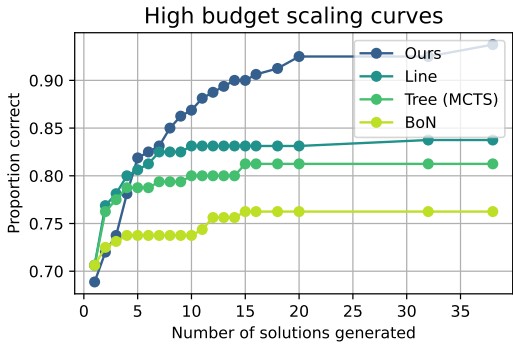

Figure 18: **High budget scaling performance for our method**. Proportion of problems where the correct solution was discovered by our method on HumanEval with `gpt-3.5-turbo0613` with 40 iterations. Our method continues to show great improvement up until 20 solutions generated while the performance of other methods plateau. Other methods do not catch up with SFS in terms of performance even with more inference scaling.

# E    PRIOR WORKS BENCHMARK

In the LATS experimental setup, in each iteration after a solution is generated, the algorithm is allowed to check if the new solution is correct on the ground truth tests before proceeding (Zhou et al., 2024). We show the performance of our method in the same setup here, with the same solution budget. The performance Pass@1 numbers are adapted from the LATS paper (Zhou et al., 2024). We see in Table 9 that given the same setup, our method still achieves higher performance.

Table 9: **Comparison to prior works**. We report Pass@1 performance with GPT-3.5.

| Method / Benchmark | HumanEval | MBPP |
|---|---|---|
| CoT  (Wei et al., 2022) | 46.9 | 54.9 |
| ReAct  (Yao et al., 2022) | 56.9 | 67.0 |
| Reflexion  (Shinn et al., 2023) | 68.1 | 70.0 |
| ToT  (Yao et al., 2023) | 54.4 | 65.8 |
| RAP  (Hao et al., 2023) | 63.1 | 71.4 |
| LATS  (Zhou et al., 2024) | 83.8 | 81.1 |
| Self-repair  (Olausson et al., 2023b) | 90.5 | 79.1 |
| Ours (SFS) | **93.3** | **91.2** |

# F    ABLATION ON MODELS

In this section, we provide a detailed ablation study on the performance of different base language models, focusing on metrics relevant to the HumanEval benchmark. Table 10 displays the results of applying various search methods across three models: `gpt-3.5`, `gpt-4o-mini`, and `gpt-4o`. For each model, we report key performance indicators such as pass@1, pass@any, BERT similarity, and validation score, along with the number of iterations (inclusive and exclusive). Furthermore, Table 11 provides an extended comparison of metrics, including similarity measures like TF-IDF and Levenshtein, alongside error rates (false positives/negatives) and true classification rates, helping to evaluate model behavior in more detail across different performance aspects.

Table 10: **Ablation on models.** We run search methods for 10 iterations each on HumanEval

| Model | pass@1 | pass@any | BERT sim. | val. score | iters. (incl) | iters. (excl) |
|-------|--------|----------|-----------|------------|---------------|---------------|
| gpt-3.5 | 82.5% | 89.0% | 0.9945 | 0.813 | 1.68 | 5.06 |
| gpt-4o-mini | 89.4% | 93.8% | 0.9876 | 0.845 | 0.84 | 5.83 |
| gpt-4o | 90.6% | 95.6% | 0.9915 | 0.894 | 0.68 | 4.95 |

Table 11: Performance metrics for different base LLM models on HumanEval. We ran `gpt-3.5-turbo-0613` for 10 iterations for each model.

| Metric | gpt-3.5 | gpt-4o-mini | gpt-4o |
|--------|---------|-------------|--------|
| Pass@Any | 89.0% | 93.8% | 95.6% |
| Pass@1 | 82.5% | 89.4% | 90.6% |
| Validation score | 0.813 | 0.845 | 0.894 |
| BERT sim. | 0.9945 | 0.9876 | 0.9915 |
| TF-IDF sim. | 0.743 | 0.643 | 0.691 |
| Levenshtein sim. | 0.721 | 0.631 | 0.665 |
| Token seq. sim. | 0.765 | 0.719 | 0.705 |
| False positive rate | 6.25% | 4.38% | 4.38% |
| False negative rate | 27.50% | 33.13% | 17.50% |
| True positive rate | 55.00% | 56.25% | 73.13% |
| True negative rate | 11.25% | 6.25% | 5.00% |
| Iters. (incl) | 1.68 | 0.84 | 0.68 |
| Iters. (excl) | 5.06 | 5.83 | 4.95 |

## F.1    INFERENCE VS TRAINING TRADEOFF

The debate between investing resources in training stronger models versus developing effective inference-time methods reflects a fundamental tradeoff in the field of machine learning. Training-based improvements often involve scaling up model size, enhancing training datasets, or incorporating domain-specific fine-tuning. While these approaches can lead to substantial performance gains, they come with high computational costs and limited adaptability post-training. Conversely, test-time methods like Scattered Forest Search (SFS) focus on maximizing the utility of pre-trained models, enabling efficient exploration and improved accuracy without additional training.

Our experiments vividly illustrate this tradeoff. For example, as shown in 14, stronger models such as GPT-4o consistently outperform weaker models like GPT-3.5. However, we also observe that weaker models benefit disproportionately more from inference scaling, highlighting that test-time optimization can be particularly valuable for models that are not extensively trained. This suggests that test-time methods can serve as a complement to training improvements, especially in resource-constrained settings.

While training stronger models remains a cornerstone of progress, inference-time techniques like SFS address challenges that training alone cannot solve. For instance:

- **Adaptability:** Test-time methods allow for domain-specific optimizations without requiring expensive retraining.

- **Exploration vs. Exploitation:** Techniques like SFS improve solution diversity and robustness, addressing issues such as overfitting or lack of generalization in pre-trained models.

- **Practical Use Cases:** Many real-world applications rely on off-the-shelf LLMs where retraining is infeasible. In such cases, enhancing inference-time performance becomes the primary lever for improvement.

Our work also reveals intriguing interaction effects between training and inference scaling. For example:

- **Better Models, Marginal Gains:** While stronger models perform better overall, the marginal benefits of inference scaling decrease as training quality improves. This is because well-trained models already exhibit diverse and high-quality outputs, reducing the need for additional exploration.

- **Weaker Models, Larger Gains:** Conversely, weaker models benefit substantially from techniques like SFS, which compensate for their limited training by expanding the search space and leveraging execution feedback to refine solutions.

These findings highlight that the importance of inference scaling is inversely proportional to the strength of the underlying model, creating an interesting trade-off that could guide future research in balancing the two approaches.

# G  PERFORMANCE METRICS FOR DIFFERENT DATASETS

We display detailed performance metrics for MBPP (Table 13), HumanEval (Table 12), CodeContests (Table 16), Leetcode (Table 14), and APPS (Table 15).

The performance metrics across various datasets, including MBPP, HumanEval, CodeContests, Leetcode, and APPS, offer a comprehensive comparison of different search methods used to evaluate large language model (LLM) performance. Each table highlights the results of different strategies, such as Line, Tree-based search (MCTS), Best of N, and our proposed method. Key metrics, such as Pass@1, Pass@Any, validation score, and various similarity metrics (e.g., BERT, TF-IDF, and Levenshtein), illustrate the effectiveness of each search method in terms of accuracy and relevance.

Additionally, false positive and false negative rates provide insight into the errors made by each approach, while true positive and negative rates reflect the precision and recall of each method. These metrics give a holistic view of model performance, especially in scenarios involving iterative search, with the number of iterations (both inclusive and exclusive) further contextualizing the computational complexity of each method.

Table 12: Performance metrics for different search methods on HumanEval. We ran `gpt-3.5-turbo-0613` for 10 iterations for each method.

| Metric | Line | Tree (MCTS) | Best of N | Ours |
|---|---|---|---|---|
| Pass@Any | 83.1% | 76.9% | 75.6% | 89.0% |
| Pass@1 | 68.1% | 75.6% | 73.8% | 82.5% |
| Validation score | 0.795 | 0.827 | 0.774 | 0.813 |
| BERT sim. | 0.9992 | 0.9998 | 0.9983 | 0.9945 |
| TF-IDF sim. | 0.495 | 0.502 | 0.936 | 0.743 |
| Levenshtein sim. | 0.480 | 0.496 | 0.925 | 0.721 |
| Token seq. sim. | 0.476 | 0.500 | 0.962 | 0.765 |
| False positive rate | 8.1% | 5.0% | 3.8% | 6.3% |
| False negative rate | 16.9% | 23.1% | 28.1% | 27.5% |
| True positive rate | 51.3% | 52.5% | 45.6% | 55.0% |
| True negative rate | 23.8% | 19.4% | 22.5% | 11.3% |
| Iters. (incl) | 2.09 | 2.61 | 2.59 | 1.68 |
| Iters. (excl) | 7.13 | 8.87 | 9.00 | 5.06 |

Table 13: Performance metrics for different search methods on MBPP. We ran `gpt-3.5-turbo-0613` for 10 iterations for each method.

| Metric | Line | Tree (MCTS) | Best of N | Ours |
|---|---|---|---|---|
| Pass@Any | 82.9% | 79.6% | 77.3% | 86.1% |
| Pass@1 | 73.6% | 76.1% | 76.6% | 78.3% |
| Validation score | 0.784 | 0.774 | 0.729 | 0.729 |
| BERT sim. | 0.9991 | 0.9998 | 0.9993 | 0.9956 |
| TF-IDF sim. | 0.473 | 0.521 | 0.960 | 0.747 |
| Levenshtein sim. | 0.455 | 0.519 | 0.956 | 0.759 |
| Token seq. sim. | 0.450 | 0.520 | 0.974 | 0.785 |
| False positive rate | 7.6% | 5.8% | 6.0% | 6.5% |
| False negative rate | 18.9% | 26.2% | 33.0% | 21.7% |
| True positive rate | 54.7% | 49.9% | 43.6% | 56.7% |
| True negative rate | 18.9% | 18.1% | 17.4% | 15.1% |
| Iters. (incl) | 1.91 | 2.29 | 2.36 | 1.91 |
| Iters. (excl) | 7.35 | 9.20 | 9.20 | 6.85 |

Table 14: Performance metrics for different search methods on Leetcode. We ran `gpt-3.5-turbo-0613` for 10 iterations for each method.

| Metric | Line | Tree (MCTS) | Best of N | Ours |
|---|---|---|---|---|
| Pass@Any | 33.3% | 33.9% | 33.9% | 39.4% |
| Pass@1 | 28.9% | 33.3% | 33.3% | 36.7% |
| Validation score | 0.352 | 0.354 | 0.372 | 0.387 |
| BERT sim. | 0.9933 | 0.9998 | 0.9936 | 0.9961 |
| TF-IDF sim. | 0.840 | 0.934 | 0.889 | 0.589 |
| Levenshtein sim. | 0.813 | 0.926 | 0.902 | 0.694 |
| Token seq. sim. | 0.818 | 0.925 | 0.897 | 0.652 |
| False positive rate | 1.1% | 0.6% | 0.6% | 5.0% |
| False negative rate | 12.2% | 27.2% | 26.7% | 24.4% |
| True positive rate | 8.3% | 6.1% | 6.7% | 9.4% |
| True negative rate | 78.3% | 66.1% | 66.1% | 61.1% |
| Iters. (incl) | 6.78 | 7.31 | 6.74 | 6.42 |
| Iters. (excl) | 9.54 | 10.44 | 9.63 | 8.75 |

Table 15: Performance metrics for different search methods on APPS. We ran `gpt-3.5-turbo-0613` for 10 iterations for each method.

| Metric | Line | Tree (MCTS) | Best of N | Ours |
|---|---|---|---|---|
| Pass@Any | 22.0% | 21.5% | 23.0% | 32.5% |
| Pass@1 | 14.5% | 18.0% | 19.5% | 20.5% |
| Validation score | 0.232 | 0.211 | 0.205 | 0.232 |
| BERT sim. | 0.9886 | 0.9997 | 0.9984 | 0.9946 |
| TF-IDF sim. | 0.837 | 0.943 | 0.825 | 0.523 |
| Levenshtein sim. | 0.817 | 0.936 | 0.843 | 0.626 |
| Token seq. sim. | 0.813 | 0.933 | 0.840 | 0.567 |
| False positive rate | 6.5% | 3.0% | 3.0% | 7.0% |
| False negative rate | 7.5% | 17.5% | 18.5% | 16.0% |
| True positive rate | 2.0% | 1.5% | 2.0% | 2.5% |
| True negative rate | 84.0% | 78.0% | 76.5% | 74.5% |
| Iters. (incl) | 7.93 | 8.66 | 7.82 | 7.30 |
| Iters. (excl) | 9.61 | 10.75 | 9.65 | 9.18 |

## H  TECHNIQUE ABLATION

In this section, we present the results of an ablation study to assess the impact of different components of our method on performance, as shown in Table 17. We evaluate four variations: the full method with all components enabled ("Everything"), and three ablations where we remove key techniques—SCATTERING, FORESTING, and SCOUTING —one at a time.

The ablation results highlight how critical each technique is for achieving high performance. Removing SCATTERING leads to a noticeable drop in Pass@Any, decreasing from 89.0% to 78.1%, while removing FORESTING has a milder effect, with Pass@Any only falling to 86.3%. Notably, FORESTING removal significantly affects similarity scores, with a drop in TF-IDF similarity from 0.743 to 0.419. Similarly, removing SCOUTING results in a slight reduction in performance metrics, but Pass@1 remains comparable to the full method, indicating its robustness.

Table 16: Performance metrics for different search methods on Codecontests. We ran `gpt-3.5-turbo-0613` for 10 iterations for each method.

| Metric | Line | Tree (MCTS) | Best of N | Ours |
|---|---|---|---|---|
| Pass@Any | 3.0% | 2.4% | 3.0% | 6.1% |
| Pass@1 | 1.21% | 2.42% | 1.82% | 4.24% |
| Validation score | 0.091 | 0.078 | 0.051 | 0.086 |
| BERT sim. | 0.9838 | 0.9997 | 0.9980 | 0.9950 |
| TF-IDF sim. | 0.871 | 0.980 | 0.770 | 0.502 |
| Levenshtein sim. | 0.856 | 0.981 | 0.800 | 0.603 |
| Token seq. sim. | 0.846 | 0.982 | 0.766 | 0.502 |
| False positive rate | 2.2% | 1.2% | 0.6% | 1.2% |
| False negative rate | 0.7% | 2.4% | 1.8% | 1.2% |
| True positive rate | 0.0% | 0.0% | 0.0% | 0.0% |
| True negative rate | 97.0% | 96.4% | 97.6% | 97.6% |
| Iters. (incl) | 9.74 | 10.73 | 9.75 | 9.53 |
| Iters. (excl) | 9.89 | 11.00 | 9.93 | 9.83 |

Table 17: Performance metrics for different ablations on techniques for our method. We ran `gpt-3.5-turbo-0613` for 10 iterations for each ablation on HumanEval.

| Metric | Everything | No SCATTERING | No FORESTING | No SCOUTING |
|---|---|---|---|---|
| Pass@Any | 89.0% | 78.1% | 86.3% | 86.3% |
| Pass@1 | 82.5% | 75.6% | 79.4% | 81.9% |
| Validation score | 0.813 | 0.802 | 0.817 | 0.792 |
| BERT sim. | 0.9945 | 0.9982 | 0.9982 | 0.9942 |
| TF-IDF sim. | 0.743 | 0.926 | 0.419 | 0.740 |
| Levenshtein sim. | 0.721 | 0.911 | 0.427 | 0.729 |
| Token seq. sim. | 0.765 | 0.945 | 0.409 | 0.756 |
| False positive rate | 0.063 | 0.063 | 0.050 | 0.063 |
| False negative rate | 0.275 | 0.244 | 0.244 | 0.281 |
| True positive rate | 0.550 | 0.513 | 0.550 | 0.538 |
| True negative rate | 0.113 | 0.181 | 0.156 | 0.119 |
| Iters. (incl) | 1.68 | 2.43 | 2.05 | 2.12 |
| Iters. (excl) | 5.06 | 8.82 | 6.56 | 6.05 |

# I VERIFIER ACCURACY AND GROUND-TRUTH VALIDATION TESTS

## I.1 VALIDATION TEST GENERATION

In our methodology, validation tests are automatically generated to evaluate the correctness of code solutions. The LLM is instructed to create Python unit tests in the form of `assert` statements. This process involves prompting the model with a high-level description of the function or module and specifying that the tests should be concise, diverse, and verifiable. By leveraging the LLM's capability to understand code semantics, the generated tests help guide the search for correct solutions and provide immediate feedback during the refinement process.

To illustrate, consider the task of generating validation tests for a function `greatest_common_divisor(a, b)`. The LLM receives a prompt to write the function and is instructed to produce unit tests in the form of `assert` statements:

> **Example Code Generation Prompt, Solution, and Tests**
>
> **Prompt:** Write a function `greatest_common_divisor(a, b)` that returns the GCD of two integers `a` and `b`.
>
> **Generated validation tests:**
> ```
> assert(greatest_common_divisor(3, 5) == 1)
> assert(greatest_common_divisor(25, 15) == 5)
> assert(greatest_common_divisor(0, 3) == 3)
> ```

The LLM follows a specific prompt instruction to generate these tests, ensuring comprehensive validation of the function's behavior. The prompt is structured as follows:

> **Test Generation Prompt**
>
> ```
> You are an AI coding assistant that can write unique,
> diverse, and comprehensive unit tests for Python objects
> given the description of the object.  The format of test
> cases should be:
> ```python
> assert function_name(input_1) == expected_output_1, "Test
> case 1 description"
> assert function_name(input_2) == expected_output_2, "Test
> case 2 description"
> ```
> DO NOT use pytest or unittest frameworks for this task.
> Stick to small inputs that you can easily verify the output
> for.
> ```

This approach ensures the generation of validation tests that are both human-readable and executable, enabling a robust evaluation of proposed solutions. By iterating through test generation, feedback, and solution refinement, the process aligns closely with real-world software development practices and ensures that the generated solutions are correct and efficient.

## I.2 TEST ACCURACY EXPERIMENTS

In this section, we assess the impact of incorporating ground-truth validation tests on verifier accuracy, as shown in Table 18. We compare three configurations: no ground-truth tests ("None"), using 3 ground-truth tests, and utilizing all available tests. These variations allow us to explore how different amounts of external validation influence the performance of our verifier.

The results demonstrate a clear improvement in Pass@1 and validation scores when ground-truth tests are introduced. With no external tests, the Pass@1 rate is 82.5%, which increases to 87.2% when 3 tests are used and further to 89.0% with all tests. Similarly, the validation score improves from 0.813 to 0.862 and 0.864, respectively. These enhancements reflect the verifier's increased accuracy when more reliable validation signals are available.

Table 18: Performance metrics when given different amounts of ground truth tests. 6 self-generated tests were used for validation in 'None'. We ran `gpt-3.5-turbo-0613` for 10 iterations for each setting.

| Metric | None | 3 tests | All tests |
|---|---|---|---|
| Pass@Any | 89.0% | 90.2% | 90.2% |
| Pass@1 | 82.5% | 87.2% | 89.0% |
| Validation score | 0.813 | 0.862 | 0.864 |
| BERT sim. | 0.9945 | 0.9952 | 0.9949 |
| TF-IDF sim. | 0.743 | 0.767 | 0.762 |
| Levenshtein sim. | 0.721 | 0.739 | 0.729 |
| Token seq. sim. | 0.765 | 0.758 | 0.757 |
| False positive rate | 6.3% | 9.8% | 1.8% |
| False negative rate | 27.5% | 2.4% | 3.7% |
| True positive rate | 55.0% | 78.0% | 85.4% |
| True negative rate | 11.3% | 9.8% | 9.1% |
| Iters. (incl) | 1.68 | 2.34 | 2.04 |
| Iters. (excl) | 5.06 | 6.86 | 5.88 |

## J   SEED SCATTERING THEME

We calculate the validation score as shown below:

$$\text{Mean validation score} = \frac{1}{|\mathcal{X}|} \sum_{\langle \boldsymbol{p}, \boldsymbol{H} \rangle \in \mathcal{X}} \frac{1}{|\mathcal{S}_{\boldsymbol{p}}|} \sum_{\boldsymbol{s} \in \mathcal{S}_{\boldsymbol{p}}} \text{proportion of validation tests passed}(\boldsymbol{s}) \quad (12)$$

We explore four themes: "None", "Jabberwocky", "Style", and "Role" to assess their impact on various evaluation metrics, including Pass@1, validation score, and similarity measures (BERT, TF-IDF, Levenshtein, and token sequence). Additionally, we provide false positive and negative rates, as well as true positive and negative rates, to further analyze the effects of the seed themes on performance in Table 19.

We also present examples of the SCATTERING seed instructions used for each theme and the metaprompt used to generate those instructions.

Table 19: Performance Metrics for varies seed instruction themes.

| Seed instruction theme | None | Jabberwocky | Style | Role |
|---|---|---|---|---|
| Pass@1 | 72.50% | 74.38% | 79.38% | 81.88% |
| Validation score | 0.7786 | 0.7658 | 0.7548 | 0.7649 |
| BERT similarity | 0.9976 | 0.9944 | 0.9929 | 0.9957 |
| TF-IDF similarity | 0.9013 | 0.7559 | 0.6826 | 0.7734 |
| Levenshtein similarity | 0.8971 | 0.7749 | 0.7119 | 0.7907 |
| Token sequence similarity | 0.9361 | 0.8444 | 0.7504 | 0.8323 |
| False positive rate | 0.0500 | 0.0438 | 0.0500 | 0.0438 |
| False negative rate | 0.3000 | 0.3063 | 0.3500 | 0.3438 |
| True positive rate | 0.4250 | 0.4375 | 0.4438 | 0.4750 |
| True negative rate | 0.2250 | 0.2125 | 0.1563 | 0.1375 |
| Pass@Any | 75.00% | 81.88% | 88.13% | 87.50% |

---

**Metaprompt: Roles of a Software Engineer**

What are different possible roles that a software engineer can have, and what are the characteristics of each role?

---

You are a problem solver. You are analytical, logical, detail-oriented. You thrive on tackling complex problems and finding efficient solutions, enjoy the challenge of debugging and often see issues as puzzles to be solved, and are methodical in your approach and persistent in your efforts to overcome obstacles.

You are an innovator. You are creative, visionary, adaptable. You are always looking for new ways to apply technology. You are not just interested in how things work but also in how they can be improved or transformed. You enjoy pioneering new techniques and technologies and are comfortable with experimentation and risk-taking.

You are a communicator. You are interpersonal, collaborative, empathetic. You excel in environments where teamwork and collaboration are key. You are skilled at explaining complex technical details in simpler terms and bridging the gap between technical teams and non-technical stakeholders. You value relationships and work well in roles that require negotiation and coordination.

Metaprompt: Instructions for Writing Code in Different Styles

What are ten different possible instructions you can give to a software engineer before they write code, instructing them to write code in three different styles.

Write the code in a highly modular way, breaking down functionality into small, reusable components. Each function or class should have a single responsibility, and avoid large monolithic structures.

Use an object-oriented approach where each concept is modeled as a class. Leverage inheritance, encapsulation, and polymorphism to create a flexible, scalable design.

Write the code in a functional programming style, avoiding mutable state and side effects. Use pure functions, higher-order functions, and recursion where appropriate.

Metaprompt: Jabberwocky Poem by Lewis Carroll

Recite the poem "Jabberwocky" by Lewis Carroll.

'Twas brillig, and the slithy toves. Did gyre and gimble in the wabe:

All mimsy were the borogoves, And the mome raths outgrabe.

Beware the Jabberwock, my son! The jaws that bite, the claws that catch!

## K    WHY SCATTERING IS USEFUL FOR SOLUTION-LEVEL TREE SEARCH

Monte Carlo Tree Search (MCTS) is a powerful algorithm used to navigate large decision spaces by balancing exploration and exploitation. At each step, MCTS selects actions based on a combination of their estimated value (exploitation) and their potential for uncovering better outcomes (exploration), typically guided by metrics such as the Upper Confidence Bound for Trees (UCT). The algorithm works by iteratively expanding the most promising nodes in the search tree, simulating outcomes, and backpropagating the results to refine its decision-making process. This iterative cycle helps MCTS focus on regions of the search space that are likely to yield optimal solutions while still dedicating some effort to exploring unvisited nodes.

The main challenge with using MCTS in our setting arises from the mechanism through which new solutions are proposed by the Large Language Model (LLM). While MCTS encourages exploration by prioritizing less-visited nodes, the diversity of exploration is fundamentally constrained by the similarity of the proposed solutions. Even if MCTS efficiently navigates the tree, if the solutions generated by the LLM are inherently similar, the result is limited exploration of the broader solution space.

This issue becomes even more pronounced in scenarios with large action spaces, such as our setting, where the action space encompasses all possible code solutions within a given length. Traditional MCTS struggles in such settings because effective exploration and exploitation require the expansion of numerous unvisited actions. Since it is computationally infeasible to exhaustively explore all possible actions, selective expansion becomes critical. Without mechanisms to enhance the diversity of these selected actions, the exploration remains suboptimal.

### K.1    TOKEN-LEVEL VS. SOLUTION-LEVEL TREE SEARCH

A significant distinction between our approach and prior works lies in the level at which MCTS operates. In traditional applications, MCTS is often employed at the token level, where it explores possible next tokens during the generation of a single solution. At this level, the LLM's log-probabilities can be used to select the top $k$ tokens, which are typically diverse enough to lead to varied outcomes. Token-level MCTS effectively balances exploration and exploitation due to the inherent variability in token selection.

In contrast, our method applies MCTS at the solution level, where each action corresponds to a complete code solution rather than individual tokens. While this approach enables broader exploration of high-level solutions, it introduces new challenges. Simply selecting the first $k$ responses generated by the LLM often results in highly similar solutions, as the LLM tends to produce consistent outputs given the same prompt. Consequently, the lack of diversity in the selected actions undermines the effectiveness of MCTS in exploring the solution space, even when it prioritizes less-visited nodes.

### K.2    ROLE OF SCATTERING FOR SOLUTION-LEVEL TREE SEARCH

To address these limitations, we introduce the Scattering technique, which explicitly diversifies the prompts used for solution generation. By incorporating varied seed themes or directions into the prompts, Scattering ensures that the LLM generates a broader range of candidate solutions. This increased diversity enhances the exploratory capabilities of MCTS, enabling it to better navigate the solution space.

Experimental results in Section 3.5 demonstrate the effectiveness of Scattering. As shown in Table 1, solutions generated without Scattering (seed theme=None) exhibit high similarity across all metrics, leading to poorer performance. In contrast, using Scattering significantly reduces the similarity of generated solutions and improves performance metrics, as depicted in Figure 11. This improvement highlights the critical role of Scattering in enabling MCTS to effectively explore large and complex action spaces.

These results underline the importance of fostering diversity in proposed actions to unlock the full potential of MCTS in large and complex action spaces, further validating the utility of the Scattering technique.

# L  ADDITIONAL EXPERIMENTS ON APPS COMPETITION-LEVEL PROBLEMS

To further evaluate the performance of our method, we conducted additional experiments on the APPS competition-level problems using ground-truth test cases. We report classic pass@k metrics as well as additional metrics designed to highlight performance variations under different resource budgets. Specifically, we measure:

- **Pass@10Req**: The proportion of problems solved within a budget of 10 LLM requests per method.
- **Pass@ntok**: The proportion of problems solved under a specified output token budget ($n$).
- **Pass@5 and Pass@10**: Standard metrics measuring the success rate when $k$ is the number of submissions allowed.

Table 20 compares the performance of different methods under these constraints using GPT-3.5-turbo. Our approach outperforms the baselines across all evaluation criteria, demonstrating its robustness and efficiency in constrained settings.

Table 20: Comparison of Methods on APPS Competition-Level Problems (Ground-Truth Test Cases, GPT-3.5-turbo)

| Method | Pass@10Req | Pass@5 | Pass@10 | Pass@1000tok | Pass@2000tok |
|---|---|---|---|---|---|
| Ours | 16.1% | 14.9% | 19.7% | 14.4% | 16.7% |
| No Scattering | 10.2% | 9.2% | 10.5% | 9.1% | 10.4% |
| No Foresting | 12.9% | 12.6% | 13.0% | 12.2% | 12.7% |
| No Scouting | 14.9% | 14.7% | 16.8% | 12.1% | 15.2% |
| Line (Reflexion) (Shinn et al., 2023) | 7.5% | 7.2% | 7.5% | 7.3% | 7.5% |
| Best of N (BoN) (Li et al., 2022) | 11.0% | 10.8% | 11.0% | 10.8% | 11.0% |
| FunSearch (Romera-Paredes et al., 2024) | 9.9% | 9.2% | 9.9% | 9.0% | 9.9% |
| REx (Tang et al., 2024) | 13.5% | 12.1% | 13.5% | 12.2% | 13.5% |
| Self-repair (Olausson et al., 2023b) | 9.3% | 8.6% | 10.7% | 7.6% | 10.1% |
| Tree (LATS) (Zhou et al., 2024) | 9.0% | 8.5% | 9.0% | 8.2% | 8.9% |
| Tree with PUCT | 9.0% | 8.9% | 9.0% | 8.8% | 9.0% |

Our method achieves superior performance across all metrics, demonstrating robustness and scalability under constrained settings. The removal of each core component—Scattering, Foresting, and Scouting—results in significant performance degradation, underscoring their necessity in achieving strong results. Among baseline methods, the REx approach performs best, followed by Best-of-N (BoN), showing the effectiveness of strong baseline search methods.

## M    THEORY ILLUSTRATION

We can imagine each code generation is a node in the search graph, and generating a new code solution given the current code and the feedback (test cases) can be seen as a transition from one node to another stochastically. The problem with direct generation of code is that they are highly similar, so the search process can be "trapped" in a local cluster where all nodes within have similar performance. The concentrated nature of the generated solutions $s$ might arise from the limited diversity inherent in the post-training objectives (Ouyang et al., 2022; Rafailov et al., 2023) commonly employed to train large language models (LLMs) as instruction-following chatbots. These objectives often prioritize optimizing the model to produce a single, correct answer. Consequently, repeated sampling from the model tends to yield highly similar outputs, offering minimal benefit from additional inference-time computation. Specifically, the whole solution space can be imagined to consist of several such node clusters. The text-based improvement direction, based on what we observed, can help prompt highly different solutions if the improvement direction is different. This can be seen as a direct edge connecting different node clusters. Such edges do not exist without text-based improvement direction. This is what Equations 3 and 4 try to establish.

Figure 19: **Illustration of Diverse Code Generation.** Left: Direct code generation explores solutions in a local cluster, leading to limited diversity. Right: Text-based improvement directions create transitions between clusters, enabling diverse solution exploration.

**Direct code generation**

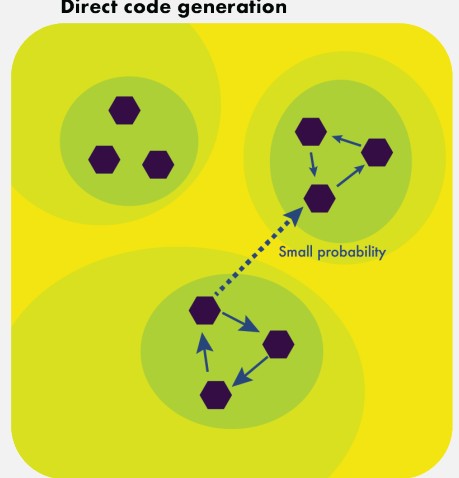

Direct code generation is trapped within a single cluster of similar solutions.

**Text guidance (from one point)**

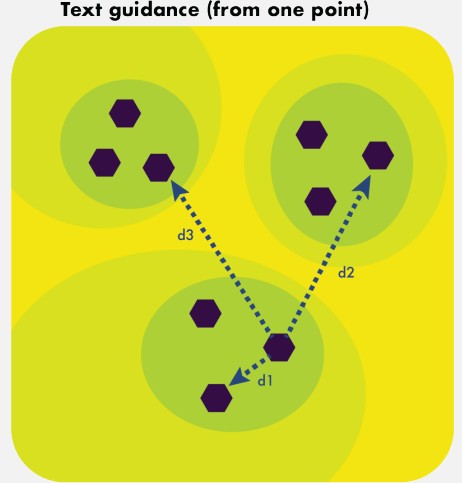

Text-based improvement introduces cross-cluster transitions, increasing solution diversity and escaping local optima.

# N    MCTS SELECTION POLICY

We evaluate two Monte Carlo Tree Search (MCTS) selection policies: PUCT (Predictor Upper Confidence Bounds for Trees) and UCT (Upper Confidence Bounds for Trees). Both policies aim to balance exploration and exploitation during search, dynamically selecting promising directions for exploration.

## N.1    SELECTION POLICIES: PUCT

PUCT extends UCT (eq. 1) by incorporating a policy prediction term $\pi(s, d)$, which biases exploration based on a prior probability distribution:

$$\text{P-UCT}(s, d) = \widehat{Q}(s, d) + \beta(s) \cdot P_{\text{Transformer}}(d|s) \cdot \sqrt{\frac{\log\left(n(s)\right)}{1 + n(s, d)}}, \tag{13}$$

where:

- $\widehat{Q}(s, d)$ is the estimated $q$-value for state $s$ and direction $d$,
- $P_{\text{Transformer}}(d|s)$ is the probability of taking direction $d$ given the partial program $s$, as predicted by the LLM,
- $n(s) = \sum_b n(s, b)$ is the total number of visits to state $s$,
- $n(s, d)$ is the number of visits to direction $d$ in state $s$,
- $\beta(s)$ is the weight for exploration, defined as:

$$\beta(s) = \log\left(\frac{n(s) + c_{\text{base}} + 1}{c_{\text{base}}}\right) + c, \tag{14}$$

where $c_{\text{base}}$ and $c$ are hyperparameters that control the exploration-exploitation balance.

The performance of PUCT heavily relies on the quality of $\pi(s, d)$, which in our case is derived from the cumulative token probabilities of the underlying language model (LLM). We use the same formula as in (Zhang et al., 2023d).

## N.2    WHY PUCT AND UCT PERFORM SIMILARLY IN OUR SETTING

Our experiments reveal that PUCT and UCT achieve similar performance levels when applied to MCTS with self-feeding search (SFS). As shown in Fig. 22 and Table 21, PUCT depends on the quality of the policy predictor, which is based on the cumulative token probabilities output by the LLM. However, in our specific setting, these probabilities do not provide strong guidance for policy-based exploration. This limitation arises because:

1. Token-level predictions are not aligned with global solution quality: The cumulative probabilities from the LLM are optimized for next-token prediction, rather than for evaluating the overall quality of solutions at different nodes in the search tree.

2. Sparse alignment between token probabilities and decision-making: For many problems in HumanEval, the token-level likelihoods fail to correlate well with successful code completions, leading to suboptimal guidance for PUCT.

3. Robustness of UCT in the absence of strong priors: UCT relies purely on the empirical $q$-value estimates and visit counts, which are updated during the MCTS process. This allows UCT to perform well even without high-quality priors.

Overall, while PUCT offers theoretical advantages when high-quality priors are available, its dependency on the LLM's token probabilities limits its effectiveness in our setting, leading to comparable performance with UCT.

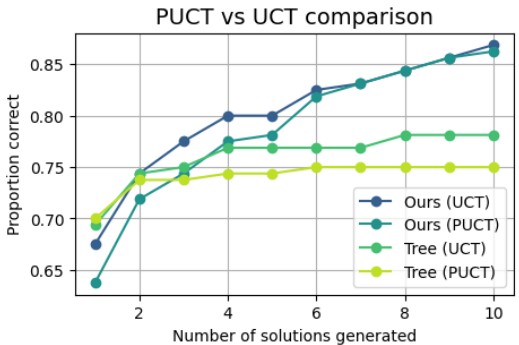

Figure 22: **PUCT vs UCT scaling performance for both SFS and tree search**. Proportion of problems where the correct solution was discovered by our method on HumanEval with `gpt-3.5-turbo0613` with 10 iterations. PUCT depends on the quality of policy predictor predictions, which are based on the cumulative token probabilities from the LLM in our case.

Table 21: **Metrics for different MCTS selection policies.** We run search methods for 10 iters. each using `gpt-3.5-turbo` on HumanEval. PUCT and UCT achieve similar performance with SFS

| Method | pass@1 | pass@any | BERT sim. | val. score | iters. (incl) | iters. (excl) |
|---|---|---|---|---|---|---|
| Tree (PUCT) | 73.8% | 75.0% | 0.9996 | 0.799 | 2.84 | 9.46 |
| Tree (UCT) | 75.6% | 76.9% | 0.9998 | 0.827 | 2.61 | 8.87 |
| Ours (PUCT) | 78.8% | 86.3% | 0.9943 | 0.788 | 2.13 | 5.88 |
| Ours (UCT) | 82.5% | 89.4% | 0.9945 | 0.813 | 1.68 | 5.06 |

