# OpenReview forum: "SFS: Smarter Code Space Search improves LLM Inference Scaling"
_ICLR.cc/2025/Conference — ICLR 2025 Poster_

### Official Review · Reviewer_4VU2 · 2024-10-22

**Soundness:** 3
**Presentation:** 2
**Contribution:** 3
**Rating:** 6
**Confidence:** 3

**Summary:**

This paper proposes SCATTERED FOREST SEARCH to encourage LLMs to produce diverse outputs. The approach incorporates scattering, foresting, and scouting techniques to strike a balance between exploration and exploitation. Extensive experiments on code generation benchmarks demonstrate its effectiveness.

**Strengths:**

* This paper is well-organized and clearly structured.
* The proposed method is sound and novel.
* The author provides a theoretical foundation for the approach.
* The experiments are thorough, and the method demonstrates strong empirical performance.

**Weaknesses:**

* The notations in Section 3 are unclear and could benefit from clarification.
  * What does $b$ represent in Eq. (1)?
  * The subscripts in lines 208-209 are incorrect.
  * In line 161, the specific direction is denoted as $d_i$, but $d_{0i}$ is used in line 206.
  * How to initialize $\hat{Q}(s,d)$?
  * How is $UCT(s,d)$ used to select directions?
  * $s$ denotes the solution in Section 3.1, but $d$ is used in line 240.
  * It would be better to add a preliminary section to introduce MCTS.
* The proposed framework introduces several layers of complexity but lacks a discussion on time complexity. How does its efficiency compare to the baseline methods?
* Minor: The figures are not clear. Please use PDF format for better readability.

**Questions:**

* Has the time or space complexity been considered for the code generation task studied in this paper?
* How do you ensure that the textual directions are orthogonal, as mentioned in line 182?
* I'm curious about the performance of the proposed framework when applied to solve combinatorial optimization problems, such as TSP.

---

> ### Author Response · Authors · 2024-11-23
> **Response 1**
>
> > The notations in Section 3 are unclear and could benefit from clarification.
>
> We thank the reviewer for this feedback and have added an additional section that explains the method in more detail, located in Appendix C. We have also improved the notation in the revised version of the paper.
>
> > What does $b$ represent in Eq. (1)?
>
> $b$ sums over improvement directions, giving us the total number of visits to that (parent) solution.
>
> > The subscripts in lines 208-209 are incorrect.
>
> We thank the reviewer for pointing these out and have fixed them in the revision accordingly.
>
> > In line 161, the specific direction is denoted as $d_i$, but $d_{0i}$ is used in line 206.
>
> We thank the reviewer for pointing these out and have fixed them in the revision accordingly. We have also changed the notation so that it is clearer.
>
> > How to initialize $\hat{Q}(s,d)?
>
> They are all initialized to 0. We have added this detail to Appendix C.
>
> > How is $UCT(s,d)$ used to select directions?
>
> We select $d^* = \arg\max_d UCT(s, d)$. This detail has also been added to Appendix C.
>
> > $s$ denotes the solution in Section 3.1, but $d$ is used in line 240.
>
> We thank the reviewer for catching the typo. It has been fixed in the revision.
>
> > It would be better to add a preliminary section to introduce MCTS.
>
> Great idea! We have included a more detailed explanation of MCTS in Appendix C.1, referenced in the methodology section per the reviewer's suggestion. Here is a high level of our explanation:
>
> Monte Carlo Tree Search (MCTS) is a powerful algorithm commonly used in decision-making and planning for games and other domains with large, complex state spaces. The core idea of MCTS is to iteratively build a search tree by simulating possible future scenarios and using the results of these simulations to guide the search towards the most promising regions of the state space. MCTS consists of four main steps in its iterative process: selection, expansion, simulation, and backpropagation. During selection, the algorithm navigates the existing search tree using a selection policy, typically based on a balance between exploration and exploitation, such as the Upper Confidence Bound (UCB) formula. Once a leaf node is reached, expansion occurs by adding one or more child nodes representing potential future states. In the simulation step, the algorithm simulates a random playout or rollout from the newly added node to estimate the outcome of the state. Finally, during backpropagation, the simulation results are propagated back through the tree, updating the value estimates and visit counts of nodes along the path. Over multiple iterations, MCTS focuses on high-value areas of the tree while still maintaining some level of exploration, making it particularly effective for tasks like game-playing, where evaluating all possible actions is computationally infeasible.

---

> ### Author Response · Authors · 2024-11-23
> **Response 2**
>
> > The proposed framework introduces several layers of complexity but lacks a discussion on time complexity. How does its efficiency compare to the baseline methods?
> Has the time or space complexity been considered for the code generation task studied in this paper?
>
> We thank the reviewer for highlighting the need for a time and space complexity analysis of our framework. To address this, we have conducted a detailed analysis and included it in Appendix C of the revised paper. Below, we summarize the key points of this analysis.
>
> **Time Complexity:**
>
> The time complexity of our Scattered Forest Search (SFS) algorithm is primarily determined by the number of iterations \(N\) and the traversal steps involved in each iteration. Similar to traditional Monte Carlo Tree Search (MCTS), SFS performs tree traversal, resulting in a worst-case time complexity of \(O(N^2)\). While this matches the complexity of tree-based search methods, SFS incorporates novel mechanisms such as scouting and scattering to enhance solution diversity and improve efficiency.
>
> For comparison:
> - **Line Search** has a time complexity of \(O(N)\), as it explores solutions sequentially without tree traversal.
> - **Beam of Nodes (BoN)** also has \(O(N)\) complexity due to maintaining and updating a fixed number of candidate solutions.
> - **Tree Search (MCTS)**, like SFS, has \(O(N^2)\) complexity due to tree traversal.
>
> SFS matches the time complexity of MCTS but leverages global insights and scattering, which improve solution quality without adding significant computational overhead.
>
> **Space Complexity:**
>
> In terms of space complexity:
> - **SFS** stores all generated solutions, resulting in \(O(N)\) space complexity. Additionally, global insights require only \(O(1)\) space, as the insights are updated iteratively and stored in a fixed-length structure.
> - **Line Search** has \(O(1)\) space complexity since it retains only the most recent solution.
> - **BoN** and **Tree Search** have \(O(N)\) space complexity, similar to SFS, as they also store candidate solutions.
>
> **Practical Considerations:**
>
> While SFS involves \(O(N^2)\) time complexity due to tree traversal, the computational bottleneck in our experiments is the solution generation process, often dominated by the cost of LLM API calls. As a result, the additional overhead introduced by tree traversal is negligible in practice.
>
> By leveraging scattering and global insights, SFS enhances solution diversity and exploration efficiency without requiring additional time or space complexity compared to baseline tree-based methods. These improvements are particularly beneficial in the code generation task studied in this paper, where solution diversity plays a crucial role in overall performance.
>
>
> > Minor: The figures are not clear. Please use PDF format for better readability.
>
> Thanks for the suggestion! We have updated the figures accordingly.
>
> > How do you ensure that the textual directions are orthogonal, as mentioned in line 182?
>
> We prompt the LLM to generate diverse and different improvement directions. Since the LLM generates all the ideas in the same response, they are more likely to be different from one another as suggested in the prompt. Note that our algorithm does not require the directions to be completely different. They just need to be diverse enough to encourage generation of more varied solutions.
>
> > I'm curious about the performance of the proposed framework when applied to solve combinatorial optimization problems, such as TSP.
>
> This is a very interesting topic, and we hope to investigate similar problems in future works!

---

> > ### Comment · Reviewer_4VU2 · 2024-11-26
> >
> > Thanks for your rebuttal, which address my concerns. I would like to keep my positive rating and to recommend acceptance. Please address other reviewers' concerns.

---

> > > ### Author Response · Authors · 2024-11-29
> > > **Thanksgiving thank you note**
> > >
> > > Thank you for your thoughtful feedback and recommendation for acceptance. We’re glad our rebuttal addressed your concerns.
> > >
> > > We’ve also carefully addressed the other reviewers’ feedback, adding clarifications and further analysis to strengthen the paper.
> > >
> > > We sincerely appreciate your support and encouragement!

---

### Official Review · Reviewer_hhUS · 2024-10-23

**Soundness:** 3
**Presentation:** 2
**Contribution:** 2
**Rating:** 5
**Confidence:** 4

**Summary:**

This paper propose a novel in-context learning workflow, which iteratively prompts a PLM to address diverse code generation tasks. Concretely, the authors proposes a reinforced tree search paradigm, termed 'scarttered forest search',  which first initializes multiple seed solutions through promping the PLM with diverse coding requirements, then proceeds the expansion of the forest search by selecting  the improvement direction with maximum UCT score. During the optimization, a global experience pool is maintained to help the exploitation. The results of the proposed searching paradigm show superiority to existing baselines, with certain diversity improvement.

**Strengths:**

Overall, the motivation of this paper is clear. The related works are properly elaborated. The experimental results are solid.

**Weaknesses:**

Although the motivation of this paper is quite clear, there are stll several issues on the rejection side:

1) The elaboration of the methodology is still far from sufficient clarity. In particular, what is the overall workflow of this paper? Can you provide a pesudocode?

2) Besides, in Section 3.1, what is the 'b' in Eq. (1)? In Section 3.3, how do you utilize the collected insights exactly? In Section 3.4, I am confused about the improvement direction 'd', which is also generated by PLM, why it is rather diverse than concentrated?

3) For me, the importance of studying how to train a LM for accurate code generation outweights the importance of studying how to prompt a PLM. This is why I denote this paper as a 'fair contribution' one.  Could the authors provide an in-depth discussion on this point to at least reveal for future readers that there is a tradeoff between the above two research direction?

4)  The experimental analysis lacks a visualization or demonstration that the proposed method really escapes the local optimal, from the results in the shown tables, I can only say that the proposed method escapes the local optima found by previous method and falls into a nes local optima now. Can the authors give a clear demonstration of what they promised in the abstract: 'Our theoretical analysis illustrates how these methods avoid local optima during optimization.'

**Questions:**

See weaknesses.

---

> ### Author Response · Authors · 2024-11-23
> **Response 1**
>
> > The elaboration of the methodology is still far from sufficient clarity. In particular, what is the overall workflow of this paper? Can you provide a pesudocode?
>
> We thank the reviewer for this suggestion. We have added more details of our method, including pseudocode and python code, in Appendix C. We give a summary below:
>
> ### Summary of the Scattered Forest Search (SFS) Algorithm
>
> ---
>
> #### Key Steps:
>
> 1. **Initialization**:
>    - **Global Insights**: A global memory $\mathcal{I}$ is initialized to store feedback and insights from the search process. This allows knowledge sharing across different branches of the search.
>    - **Forest Creation**: The algorithm generates a set of diverse seed solutions $\mathcal{S}_0 = \{\boldsymbol{s}_1, \boldsymbol{s}_2, \dots, \boldsymbol{s}_n\}$ using a language model (LM). For each seed:
>      - **Direction Generation**: A set of diverse improvement directions $\{d_1, d_2, \dots, d_k\}$ is generated using the LM, conditioned on feedback from the seed.
>      - **Q-Values**: **Q-values for all directions are initialized to 0**, i.e., $$node.qvalue(d_i) \leftarrow 0$$.
>
> 2. **Seed Selection**:
>    - The most promising seed solution is selected from the forest $\mathcal{S}$ using the Upper Confidence Bound for Trees (UCT) formula:
>      $$
>      UCT(\boldsymbol{s}, \boldsymbol{d}) = \hat{Q}(\boldsymbol{s}, \boldsymbol{d}) + c \sqrt{\frac{\ln \left(\sum_{\boldsymbol{b}} n(\boldsymbol{s}, \boldsymbol{b})\right)}{n(\boldsymbol{s}, \boldsymbol{d})}},
>      $$
>      where $\hat{Q}(\boldsymbol{s}, \boldsymbol{d})$ is the estimated reward, $n(\boldsymbol{s}, \boldsymbol{d})$ is the visit count, and $c$ controls the exploration-exploitation trade-off.
>
> 3. **Simulation**:
>    - Starting from the selected seed, a trajectory $\tau = [(\boldsymbol{s}_i, \boldsymbol{d}_i, \boldsymbol{s}_{i+1})]$ is simulated. At each step, the most promising direction $\boldsymbol{d}_i$ is selected using the UCT formula. The process continues until a leaf node is reached.
>
> 4. **Expansion**:
>    - At the leaf node, a new solution $\boldsymbol{s}'$ is generated by applying an improvement direction $\boldsymbol{d}$:
>      $$
>      \boldsymbol{s}' = \mathcal{L}(\boldsymbol{s}, \boldsymbol{d}, \text{feedback}),
>      $$
>      where the LM generates $\boldsymbol{s}'$ based on the parent solution $\boldsymbol{s}$, the direction $\boldsymbol{d}$, and validation feedback.
>    - **Reward**: The generated solution is validated, and the **reward is the proportion of validation tests passed**:
>      $$
>      \text{Reward} = \frac{\text{Number of tests passed}}{\text{Total number of tests}}.
>      $$
>
> 5. **Backpropagation**:
>    - The reward is propagated back along the trajectory $\tau$ using the backpropagation update rule:
>      $$
>      \hat{Q}(\boldsymbol{s}_i, \boldsymbol{d}_{i+1})^{(t+1)} \leftarrow (1-\alpha_n) \hat{Q}(\boldsymbol{s}_i, \boldsymbol{d}_{i+1})^{(t)} + \alpha_n \max\{\hat{Q}(\boldsymbol{s}_i, \boldsymbol{d}_{i+1})^{(t)}, \hat{Q}(\boldsymbol{s}_{i+1}, \boldsymbol{d}_{i+2})^{(t+1)}\},
>      $$
>      where $\alpha_n$ is a weighting parameter that depends on the visit count $n(\boldsymbol{s}, \boldsymbol{d})$.
>    - **Visit Count**: The visit count $n(\boldsymbol{s}, \boldsymbol{d})$ is incremented for each direction.
>
> 6. **Scouting**:
>    - Feedback from each expansion is analyzed to extract general insights. These insights are stored in the global memory $\mathcal{I}$ and reused in future searches to guide direction generation. This enhances the algorithm’s exploitation capabilities.
>
> 7. **Scattering**:
>    - After each expansion, diverse new directions $\{d_1, d_2, \dots, d_k\}$ are generated for the newly created node using the LM. This encourages exploration of different regions of the solution space, reducing the likelihood of stagnation in local optima.
>
> 8. **Final Selection**:
>    - After $N$ iterations, the best solution $s^*$ across all trees in the forest is selected. The best solution is determined based on the highest reward (i.e., the **proportion of validation tests passed**).
>
> ---
>
> #### Key Features:
> - **Q-Value Initialization**: For every direction, **Q-values are initialized to 0** and updated dynamically during backpropagation.
> - **Reward Definition**: The reward is explicitly defined as the **proportion of validation tests passed**, providing a consistent and interpretable performance metric.
> - **Diversity**: The scattering technique ensures diversity in both initial seed solutions and improvement directions, allowing the algorithm to explore a wider solution space.
> - **Global Insights**: The scouting mechanism enables the sharing of effective strategies across branches, improving the efficiency of feedback exploitation.
>
> The integration of scattering, foresting, and scouting enables the SFS algorithm to balance exploration and exploitation effectively, making it a robust method for navigating complex search spaces.

---

> ### Author Response · Authors · 2024-11-23
> **Response 2**
>
> > Besides, in Section 3.1, what is the 'b' in Eq. (1)? In Section 3.3, how do you utilize the collected insights exactly? In Section 3.4, I am confused about the improvement direction 'd', which is also generated by PLM, why it is rather diverse than concentrated?
>
> Thank you for raising these points. First, there is a typo in Eq. (1), which we have corrected in the revised version. In this equation, $b$ represents the sum over improvement directions, providing the total number of visits to a given solution. This clarification has been added to ensure the equation is fully interpretable.
>
> Regarding the collected insights in Section 3.3, these are utilized when proposing new improvement directions for exploration. Specifically, the global insights are updated each time an improvement direction is attempted. The LLM reflects on whether the improvement idea was effective based on the execution feedback, and this reflection informs future direction proposals. This iterative feedback loop allows us to focus exploration on promising regions of the solution space. We appreciate the reviewer identifying this area for clarification, and we have added a more detailed explanation of this mechanism in Appendix C.
>
> In Section 3.4, the diversity of improvement directions $d$ stems from their textual nature, which contrasts with the concentrated nature of the generated solutions $s$. The concentrated nature of the generated solutions $s$ might arise from the limited diversity inherent in the post-training objectives [1, 2] commonly employed to train large language models (LLMs) as instruction-following chatbots. These objectives often prioritize optimizing the model to produce a single, correct answer. Consequently, repeated sampling from the model tends to yield highly similar outputs, offering minimal benefit from additional inference-time computation. We thus propose to explore improvement direction 'd' expressed in natural language as a means to enhance diversity in the code search process. We empirically validated the effectiveness of this idea. As shown in Table 1, directly generating solution $s$ without using text-based directions leads to highly similar code outputs across all similarity metrics. In practice, the LLM-generated text distribution (used for $d$) tends to be more diverse than the distribution of solutions $s$. This is a key hypothesis underlying our approach, as text-based diversity enables broader exploration of the code space. To address this confusion, we have included a more detailed explanation of Section 3.4 in Appendix L.
>
> We thank the reviewer for bringing these areas to our attention, as it allowed us to further clarify and enhance the presentation of our method.
>
> [1] Long Ouyang, Jeffrey Wu, Xu Jiang, Diogo Almeida, Carroll Wainwright, Pamela Mishkin, Chong Zhang, Sandhini Agarwal, Katarina Slama, Alex Ray, et al. Training language models to follow instructions with human feedback. NeurIPS, 2022.
> [2] Rafael Rafailov, Archit Sharma, Eric Mitchell, Christopher D Manning, Stefano Ermon, and Chelsea Finn. Direct preference optimization: Your language model is secretly a reward model. NeurIPS, 2023.

---

> ### Author Response · Authors · 2024-11-23
> **Response 3**
>
> > For me, the importance of studying how to train a LM for accurate code generation outweights the importance of studying how to prompt a PLM. This is why I denote this paper as a 'fair contribution' one. Could the authors provide an in-depth discussion on this point to at least reveal for future readers that there is a tradeoff between the above two research direction?
>
> Thank you for this insightful comment. We agree that the tradeoff between training an LM for accurate code generation and improving test-time performance via prompting and search-based methods is a critical topic that warrants deeper exploration. Below, we provide a detailed discussion to address this point and contextualize our contribution:
>
> 1. **Tradeoff Between Training and Test-Time Optimization:**
>    The debate between investing resources in training stronger models versus developing effective inference-time methods reflects a fundamental tradeoff in the field of machine learning. Training-based improvements often involve scaling up model size, enhancing training datasets, or incorporating domain-specific fine-tuning. While these approaches can lead to substantial performance gains, they come with high computational costs and limited adaptability post-training. Conversely, test-time methods like Scattered Forest Search (SFS) focus on maximizing the utility of pre-trained models, enabling efficient exploration and improved accuracy without additional training.
>
>    Our experiments vividly illustrate this tradeoff. For example, as shown in **Figure 14**, stronger models such as GPT-4o consistently outperform weaker models like GPT-3.5. However, we also observe that weaker models benefit disproportionately more from inference scaling, highlighting that test-time optimization can be particularly valuable for models that are not extensively trained. This suggests that test-time methods can serve as a complement to training improvements, especially in resource-constrained settings.
>
>    More detailed statistics and discussion on how training affects inference scaling are presented in **Appendix F**, where we provide additional metrics comparing models of varying strengths. These results further underscore the interaction effects between training and inference techniques.
>
> 2. **Complementary Roles of Training and Inference Scaling:**
>    While training stronger models remains a cornerstone of progress, inference-time techniques like SFS address challenges that training alone cannot solve. For instance:
>    - **Adaptability:** Test-time methods allow for domain-specific optimizations without requiring expensive retraining.
>    - **Exploration vs. Exploitation:** Techniques like SFS improve solution diversity and robustness, addressing issues such as overfitting or lack of generalization in pre-trained models.
>    - **Practical Use Cases:** Many real-world applications rely on off-the-shelf LLMs where retraining is infeasible. In such cases, enhancing inference-time performance becomes the primary lever for improvement.
>
> 3. **Interaction Effects Between Training and Inference:**
>    Our work also reveals intriguing interaction effects between training and inference scaling. For example:
>    - **Better Models, Marginal Gains:** While stronger models perform better overall, the marginal benefits of inference scaling decrease as training quality improves. This is because well-trained models already exhibit diverse and high-quality outputs, reducing the need for additional exploration.
>    - **Weaker Models, Larger Gains:** Conversely, weaker models benefit substantially from techniques like SFS, which compensate for their limited training by expanding the search space and leveraging execution feedback to refine solutions.
>
>    These findings highlight that the importance of inference scaling is inversely proportional to the strength of the underlying model, creating an interesting tradeoff that could guide future research in balancing the two approaches.
>
> 4. **Future Research Directions:**
>    Both research directions—training and inference-time optimization—offer compelling opportunities for future exploration. For training, improving data quality, fine-tuning strategies, and incorporating code-specific pretraining objectives are promising paths. For inference, developing more sophisticated search algorithms, leveraging multi-agent interactions, or combining retrieval-augmented generation with methods like SFS could yield further advancements. Understanding the interplay between these two directions will be critical for advancing the field.
>
> We hope this expanded discussion, along with the additional statistics in **Appendix F**, provides valuable insights into the tradeoff between training and inference-time methods and contextualizes the relevance of our contribution. Thank you for encouraging us to deepen this discussion.

---

> ### Author Response · Authors · 2024-11-23
> **Response 4**
>
> > The experimental analysis lacks a visualization or demonstration that the proposed method really escapes the local optimal, from the results in the shown tables, I can only say that the proposed method escapes the local optima found by previous method and falls into a nes local optima now. Can the authors give a clear demonstration of what they promised in the abstract: 'Our theoretical analysis illustrates how these methods avoid local optima during optimization.'
>
> Thank you for this thoughtful comment. We agree that a clearer demonstration of how our method avoids local optima would strengthen the paper. To address this, we provide additional theoretical insights and a new illustration (Figure 19) to clarify this concept.
>
> The code generation process can be visualized as a search graph, where each node represents a generated code solution, and transitions between nodes correspond to generating a new solution based on the current code and feedback (e.g., test cases). Without mechanisms to enhance diversity, directly generating code solutions often results in outputs that are highly similar. This leads to the search process becoming "trapped" within a local cluster of nodes, where all solutions share similar performance. Conceptually, the entire solution space can be imagined as comprising multiple such clusters, with limited connectivity between them.
>
> Our idea is to explore improvement directions (i.e, ideas) expressed in natural language as a means to enhance diversity in the code search process. We empirically validated the effectiveness of this idea. Our approach introduces text-based improvement directions that act as a mechanism to add docoding diversity so as to "jump" across these clusters. Based on our observations, these directions enable the generation of highly diverse solutions when the textual improvement prompts differ. This can be visualized as the creation of direct edges between otherwise disconnected node clusters, allowing the search process to escape local optima and explore broader regions of the solution space. Without these text-based improvement directions, such edges are effectively absent, leaving the search constrained to a single cluster.
>
> Equations (3) and (4) formalize this concept by highlighting how the inclusion of text-based directions expands the transition kernel, thereby increasing the probability of escaping local clusters. To make this explanation more accessible, we have added Figure 19 to illustrate the impact of these transitions. Furthermore, we have included a detailed explanation in Appendix L to elaborate on this theoretical perspective and provide additional clarity.
>
> We hope this addresses the reviewer’s concern and demonstrates how our method can more likely avoid local optima, as described in the abstract. Thank you for encouraging us to improve the presentation of this critical aspect of our work.

---

> > ### Comment · Reviewer_hhUS · 2024-11-25
> > **reply**
> >
> > Thanks for the detailed response, with corresponding update in the revised paper. I will keep my score due to two reasons: a) For me, how to design comprehensive CoT patterns for training Code LLMs outweights how to prompt them, while I acknowledge the discussion of the tradeoff between them the authors have added is useful for the community. b) I fully understand the motivation and rational behind eq. (3) and (4), but for now, it only provides an oppotunity for this work to escape the local optimum when naively prompting Code LLMs. I still expect more convincing analysis results to demonstrate how nearly this work approaches the global optimum (not just empirical performance improvement against some baselines). In summary, I keep my score and wait for other reviewers' opinions.

---

### Official Review · Reviewer_zwDf · 2024-11-02

**Soundness:** 2
**Presentation:** 2
**Contribution:** 3
**Rating:** 6
**Confidence:** 2

**Summary:**

This paper discusses the search strategy with LLM-based operators for code generation. It has an interesting prompting method to increase the diversity of LLM generations: first ask LLM to generate a list of chain-of-thoughts (COTs) and then ask LLM to generate code given each chain of thought. The generated COTs tend to be more diverse due to the itemized prompting and then the codes get more diverse. It also caches the chain of thoughts to reuse on other codes. One more claimed contribution is using a forest of search trees to "encourage exploration". It compares with simple baselines and reports improved performance in a customized setting.

**Strengths:**

The itemized chain-of-thought generation prompting method is interesting. It seems to be a general, intuitive, and easy-to-implement prompting method that could potentially be applied in many other reasoning-based domains where the diversity of LLM generations is important. It also unleashed the most significant improvement in the ablation study. I would recommend the authors to test this prompting technique in more domains such as generating math proofs. It would be an interesting addition to our toolbox for LLM-based reasoning.

Code generation is an important domain to study.

**Weaknesses:**

* **Non-standard and vague experimental setting**s: This paper claims to use 6 self-generated test cases for each problem to solve. However, I couldn't find any related descriptions or details about how they generate them. The self-generated test cases are neither the contribution of this paper nor part of some public datasets/benchmarks. It is thus hard to reproduce and compare results from this paper with other methods in this domain. It is also unclear how many LLM requests/tokens they use in these experiments.
* **Missing baselines; only simplest baselines**: There are many more papers discussing the search strategy with LLM-based code refinements/generation (exploration v.s. exploitation), such as Fun-Search [1] based on evolution search and REx[2] based on bandits algorithms. There are many more popular hard-coded tree expansion policies in the field such as [3,4].
* **Missing details of the method**: There are many confusing or missing descriptions of the method such as (1) the definition of rewards/values for each action/code; (2) how to initialize Q in UCB/MCTS; (3) how to generate test cases; (4) what is the overall pipeline? how many LLM requests for each "submission"?
* **Complicated method; Unclear contributions of each component**: This paper includes many components in its method including (1) self-generated test cases, (2) itemized COT prompting, (3) UCB-based tree-expansion policy, (4) reusing COTs, (5) "foresting", and so on. It is unclear how each component contributes to the performance. I would suggest the authors perform ablation studies in a more common/standard setting, removing influences of components that are not part of this paper's contributions.


[1] Romera-Paredes, Bernardino, et al. "Mathematical discoveries from program search with large language models." Nature 625.7995 (2024): 468-475.

[2] Tang, Hao, et al. "Code Repair with LLMs gives an Exploration-Exploitation Tradeoff." arXiv preprint arXiv:2405.17503 (2024).

[3] Olausson, Theo X., et al. "Is Self-Repair a Silver Bullet for Code Generation?." The Twelfth International Conference on Learning Representations. 2023.

[4] Wang, Ruocheng, et al. "Hypothesis search: Inductive reasoning with language models." arXiv preprint arXiv:2309.05660 (2023).

**Questions:**

See the weakness for details.

* Are there clearer descriptions of the method, from the overall pipeline to each component such as the reward/value design?
* Are there results without the self-generated test cases which are neither this paper's contribution nor part of standard benchmarks?
* Are there results of more popular baselines than simple line-search?
* Are there more detailed descriptions of the experimental settings, including how many LLM requests are used?
* Are there ablation studies in a cleaner or more standard setting?

---

> ### Author Response · Authors · 2024-11-23
> **Response 1**
>
> > Non-standard and vague experimental settings: This paper claims to use 6 self-generated test cases for each problem to solve. However, I couldn't find any related descriptions or details about how they generate them. The self-generated test cases are neither the contribution of this paper nor part of some public datasets/benchmarks. It is thus hard to reproduce and compare results from this paper with other methods in this domain. It is also unclear how many LLM requests/tokens they use in these experiments.
>
> Thank you for pointing this out. We apologize for the lack of clarity and have made significant revisions to address these concerns. Below, we provide a detailed explanation of our experimental setup, including the rationale for using self-generated test cases, comparisons with settings involving ground-truth validation tests, and specifics about LLM requests and tokens.
>
> 1. **Experimental Setup and Use of Self-Generated Test Cases:**
>    In our problem setting, the solver is given only the problem prompt, not the hidden ground-truth tests used for evaluating correctness. The solver is allowed to employ any method to solve the problem, including self-generating validation tests, provided it does not access the hidden tests. This constraint ensures that the task closely mirrors real-world applications of program synthesis, where the solver must produce solutions based solely on the problem prompt $\boldsymbol{p}$. Such a setup aligns with platforms like LeetCode or conventional coding assessments, where test cases remain hidden to preserve the integrity of the evaluation process and discourage overfitting to known test sets.
>
>    This restriction is not only practical but also reflects the needs of end users in real-world scenarios. Writing comprehensive and correct unit tests is a time-consuming process, requiring substantial domain knowledge and effort. In practice, users would prefer tools that can autonomously generate correct implementations from a prompt without requiring additional test specifications or guidance. By requiring solvers to rely on self-generated validation tests or implicit reasoning to produce correct solutions, this setup encourages the development of tools capable of robust and generalized problem-solving.
>
>    By prioritizing correctness under these constraints, our framework evaluates an agent’s ability to understand the problem and synthesize accurate solutions in a realistic and user-centric manner, making it an ideal benchmark for program synthesis tasks.
>
>    Many prior works use similar setups involving self-generated validation tests during the solving process (e.g., [5–14]). However, we have now included a more detailed explanation of our problem setting in **Appendix B** and the **Problem Description** section. This ensures that future researchers can understand our problem setting better.
>
> 2. **Validation test generation process**
>     In our methodology, validation tests are automatically generated to evaluate the correctness of code solutions. The LLM is instructed to create Python unit tests in the form of \texttt{assert} statements. This process involves prompting the model with a high-level description of the function or module and specifying that the tests should be concise, diverse, and verifiable. By leveraging the LLM’s capability to understand code semantics, the generated tests help guide the search for correct solutions and provide immediate feedback during the refinement process.
>     We have included a more detailed explanation of our generation method in Appendix I.
>
>
> 3. **Experiments with Ground-Truth Validation Tests:**
>    To address concerns about reproducibility and comparisons, we conducted additional experiments using ground-truth validation tests. The results are summarized below (details provided in **Appendix I**):
>
>    | **Tests given**   | **pass@1** | **pass@any** | **BERT sim.** | **val. score** | **iters. (incl)** | **iters. (excl)** |
>    |--------------------|------------|--------------|---------------|----------------|-------------------|-------------------|
>    | No tests given     | 82.5%      | 89.0%        | 0.9945        | 0.813          | 1.68              | 5.06              |
>    | 3 tests given      | 87.2%      | 90.2%        | 0.9952        | 0.862          | 2.34              | 6.86              |
>    | All tests given    | 89.0%      | 90.2%        | 0.9949        | 0.864          | 2.04              | 5.88              |
>
>    As shown, while ground-truth validation tests significantly improve the **pass@1** rate, they do not notably affect **pass@any** rates. However, ground-truth tests increase the mean validation score, demonstrating that with a more accurate evaluation signal, our search algorithm can more effectively explore promising regions of the solution space. These results highlight both the robustness of our method to noisy validation signals and its ability to perform well even with self-generated tests.

---

> ### Author Response · Authors · 2024-11-23
> **Response 1 (continued)**
>
> 4. **Accuracy of Self-Generated Validation Tests:**
>    We further analyzed the accuracy of self-generated validation tests in **Figure 13**, which shows that these tests are accurate approximately **66.25%** of the time. Despite this noise, our method achieves comparable **pass@any** rates to settings with ground-truth validation tests (as shown in Table 8). This robustness to validation noise highlights the adaptability and reliability of our approach.
>
> 5. **Comparison to Baselines with Ground-Truth Tests:**
>    To ensure comparability with prior methods, we also evaluated our algorithm in settings where some ground-truth tests were provided. The results are reproduced below (Table 5):
>
>    | **Benchmark**          | **HumanEval** | **MBPP** |
>    |-------------------------|---------------|----------|
>    | **Tests given**         | 3             | 1        |
>    | SD (+Expl.)             | 81.1          | 74.4     |
>    | SD (+Trace)             | 80.5          | 72.6     |
>    | LDB                     | 82.9          | 76.0     |
>    | **Ours**                | **87.2**      | **91.3** |
>
>    These results show that our method significantly outperforms prior baselines in these settings.
>
> 6. **LLM API Usage and Requests:**
>    Each solution generated by our algorithm requires two API calls:
>    - **Code Generation Call:** This generates a solution based on Scattering improvement directions.
>    - **Reflection Call:** This updates global insights based on execution feedback and generates new Scattering directions for subsequent iterations.
>
>    We have added these details to **Appendix C**.
>
> 7. **Token usage**
>     Per the reviewer's suggestion, we have included a scaling chart (Figure 12, right) that shows how our method scales with token usage. We see that our method scales better and performs better than the baselines given the same token budget.
>
> We hope these clarifications and additional results have helped clarify our method. Thank you for this valuable feedback, which has greatly helped us improve the presentation and rigor of our work.

---

> ### Author Response · Authors · 2024-11-23
> **Response 2**
>
> > Missing baselines; only simplest baselines: There are many more papers discussing the search strategy with LLM-based code refinements/generation (exploration v.s. exploitation), such as Fun-Search [1] based on evolution search and REx[2] based on bandits algorithms. There are many more popular hard-coded tree expansion policies in the field such as [3,4].
>
> Thank you for highlighting these additional related works and baselines. We agree that comparing our method against a broader range of search strategies is crucial for contextualizing its contributions. In response, we have added comparisons to the suggested baselines, including FunSearch [2], REx [2], and several other prior works, in **Appendix E** and **Tables 5, 6, and 9**. Below, we summarize the results to provide a comprehensive comparison.
>
> First, in **Table 9**, we compare our method's performance on Pass@1 metrics with GPT-3.5 against a range of prior approaches, including methods like CoT, ReAct, Reflexion, ToT, RAP, LATS, Self-repair, and others. Our method (SFS) consistently achieves superior performance across both the HumanEval and MBPP benchmarks, demonstrating its effectiveness in generating high-quality code solutions.
>
> ### Table: Comparison to Prior Works
> We report Pass@1 performance with GPT-3.5.
>
> | **Method / Benchmark**                | **HumanEval** | **MBPP** |
> |---------------------------------------|---------------|----------|
> | CoT ~[@wei2022chain]                  | 46.9          | 54.9     |
> | ReAct ~[@yao2022react]                | 56.9          | 67.0     |
> | Reflexion ~[@shinn2024reflexion]      | 68.1          | 70.0     |
> | ToT ~[@yao2024tree]                   | 54.4          | 65.8     |
> | RAP ~[@hao2023reasoning]              | 63.1          | 71.4     |
> | LATS ~[@zhou2023language]             | 83.8          | 81.1     |
> | Self-repair ~[@olausson2023self]      | 90.5          | 79.1     |
> | **Ours (SFS)**                        | **93.3**      | **91.2** |
>
> In **Table 5**, we compare our method in settings where ground-truth validation tests are partially available. This aligns with baselines like SD (+Expl.), SD (+Trace), and LDB, as suggested by the reviewer. Again, our method demonstrates superior performance, particularly in MBPP, where it achieves a Pass@1 score of 91.3%.
>
> ### Table: Comparison to Prior Works with Ground Truth Tests
> We report Pass@1 performance with GPT-3.5 when a subset of ground truth tests are given.
>
> | **Benchmark**               | **HumanEval** | **MBPP** |
> |-----------------------------|---------------|----------|
> | **Tests given**             | **3**         | **1**    |
> | SD (+Expl.) ~[@chen2023teaching] | 81.1          | 74.4     |
> | SD (+Trace) ~[@chen2023teaching] | 80.5          | 72.6     |
> | LDB ~[@zhong2024ldb]             | 82.9          | 76.0     |
> | **Ours (SFS)**                   | **87.2**      | **91.3** |
>
> We also conducted comparisons with REx [2], as suggested by the reviewer. Since the REx codebase is not publicly available, we used the reported results from their paper on the APPS benchmark. On this benchmark, our method achieves a Pass@1 performance of **32.5%**, significantly outperforming REx’s reported performance of **20.2%**.
>
> In **Table 6**, we extend our comparisons to include FunSearch, as suggested. These results highlight the efficiency and diversity of our method (SFS), which discovers correct solutions faster and generates more diverse outputs. Our method outperforms FunSearch and other baselines across multiple metrics, including Pass@1, Pass@any, and solution diversity.
>
> ### Table: Metrics for Different Search Methods
> We run search methods for 10 iterations each using `gpt-3.5-turbo` on HumanEval. Our method generates more diverse solutions and discovers the correct solution faster.
>
> | **Method**      | **pass@1** | **pass@any** | **BERT sim.** | **val. score** | **iters. (incl)** | **iters. (excl)** |
> |-----------------|------------|--------------|---------------|----------------|-------------------|-------------------|
> | Line            | 68.1%      | 83.1%        | 0.9992        | 0.795          | 2.09              | 7.13              |
> | Tree (MCTS)     | 75.6%      | 76.9%        | 0.9998        | 0.827          | 2.38              | 8.09              |
> | Best of N       | 73.8%      | 75.6%        | 0.9983        | 0.774          | 2.59              | 9.00              |
> | FunSearch [2]   | 74.4%      | 75.6%        | 0.9994        | 0.815          | 3.04              | 10.36             |
> | **Ours (SFS)**  | **82.5%**  | **89.0%**    | **0.9945**    | **0.813**      | **1.67**          | **5.06**          |
>
> We appreciate the reviewer’s suggestion to include these comparisons, as they have allowed us to strengthen the evaluation of our method and demonstrate its advantages more comprehensively.

---

> ### Author Response · Authors · 2024-11-23
> **Response 3**
>
> > Missing details of the method: There are many confusing or missing descriptions of the method such as (1) the definition of rewards/values for each action/code; (2) how to initialize Q in UCB/MCTS; (3) how to generate test cases; (4) what is the overall pipeline? how many LLM requests for each "submission"?
>
> Thank you for identifying these areas where additional clarification is needed. We address each point below in detail:
>
> 1. **Definition of Rewards/Values for Each Action/Code:**
>    The reward for a given code solution is defined as the proportion of validation tests passed. These rewards are propagated during the Monte Carlo Tree Search (MCTS) to update the value $Q(s, d)$ associated with specific actions (improvement directions). This mechanism encourages the search process to explore promising solutions while avoiding poorly performing ones. By linking rewards directly to validation test outcomes, the method ensures that progress is grounded in concrete feedback on solution correctness. We have clarified this definition further in **Appendix C**.
>
> 2. **Initialization of $Q$ in UCB/MCTS:**
>    In our implementation of MCTS, the $Q$ values are initialized to $0$ for all nodes. These values are updated immediately after the first visit to a node, ensuring that the search process reflects observed outcomes as soon as data is available. This approach aligns with standard practices in MCTS and has been detailed in the revised paper for clarity.
>
> 3. **Test Case Generation:**
>    Validation tests in our framework are automatically generated by the LLM to evaluate the correctness of code solutions. The LLM is prompted to create Python unit tests in the form of `assert` statements, using a high-level description of the target function or module. To ensure robustness, the prompts specify that the tests should be concise, diverse, and verifiable. This leverages the LLM’s understanding of code semantics to produce meaningful validation tests that guide the search process and provide immediate feedback for refinement. Additional details on this method have been included in **Appendix I**.
>
> 4. **Number of LLM Requests Per Submission:**
>    Each solution submission involves two API calls:
>    - **Code Generation Call:** Generates a solution based on Scattering improvement directions.
>    - **Reflection Call:** Updates global insights using execution feedback and produces new Scattering directions for subsequent iterations.
>
>    This design ensures that both solution diversity and iterative refinement are effectively supported. We have added a detailed breakdown of these steps in **Appendix C**.
>
> 5. **Overall pipeline**
>
>  We have added more details of our method, including pseudocode and python code, in Appendix C. We give a summary of the pipeline below:

---

> ### Author Response · Authors · 2024-11-23
> **Response 3 (continued)**
>
> ### Summary of the Scattered Forest Search (SFS) Algorithm
>
> The **Scattered Forest Search (SFS)** combines optimization-inspired techniques and tree search, integrating three core components: **scattering**, **foresting**, and **scouting**. Below is a detailed summary of the algorithm's operation.
>
> ---
>
> #### Key Steps:
>
> 1. **Initialization**:
>    - **Global Insights**: A global memory $\mathcal{I}$ is initialized to store feedback and insights from the search process. This allows knowledge sharing across different branches of the search.
>    - **Forest Creation**: The algorithm generates a set of diverse seed solutions $\mathcal{S}_0 = \{\boldsymbol{s}_1, \boldsymbol{s}_2, \dots, \boldsymbol{s}_n\}$ using a language model (LM). For each seed:
>      - **Direction Generation**: A set of diverse improvement directions $\{d_1, d_2, \dots, d_k\}$ is generated using the LM, conditioned on feedback from the seed.
>      - **Q-Values**: **Q-values for all directions are initialized to 0**, i.e., $$node.qvalue(d_i) \leftarrow 0$$
>
> 2. **Seed Selection**:
>    - The most promising seed solution is selected from the forest $\mathcal{S}$ using the Upper Confidence Bound for Trees (UCT) formula:
>      $$
>      UCT(\boldsymbol{s}, \boldsymbol{d}) = \hat{Q}(\boldsymbol{s}, \boldsymbol{d}) + c \sqrt{\frac{\ln \left(\sum_{\boldsymbol{b}} n(\boldsymbol{s}, \boldsymbol{b})\right)}{n(\boldsymbol{s}, \boldsymbol{d})}},
>      $$
>      where $\hat{Q}(\boldsymbol{s}, \boldsymbol{d})$ is the estimated reward, $n(\boldsymbol{s}, \boldsymbol{d})$ is the visit count, and $c$ controls the exploration-exploitation trade-off.
>
> 3. **Simulation**:
>    - Starting from the selected seed, a trajectory $\tau = [(\boldsymbol{s}_i, \boldsymbol{d}_i, \boldsymbol{s}_{i+1})]$ is simulated. At each step, the most promising direction $\boldsymbol{d}_i$ is selected using the UCT formula. The process continues until a leaf node is reached.
>
> 4. **Expansion**:
>    - At the leaf node, a new solution $\boldsymbol{s}'$ is generated by applying an improvement direction $\boldsymbol{d}$:
>      $$
>      \boldsymbol{s}' = \mathcal{L}(\boldsymbol{s}, \boldsymbol{d}, \text{feedback}),
>      $$
>      where the LM generates $\boldsymbol{s}'$ based on the parent solution $\boldsymbol{s}$, the direction $\boldsymbol{d}$, and validation feedback.
>    - **Reward**: The generated solution is validated, and the **reward is the proportion of validation tests passed**:
>      $$
>      \text{Reward} = \frac{\text{Number of tests passed}}{\text{Total number of tests}}.
>      $$
>
> 5. **Backpropagation**:
>    - The reward is propagated back along the trajectory $\tau$ using the backpropagation update rule:
>      $$
>      \hat{Q}(\boldsymbol{s}_i, \boldsymbol{d}_{i+1})^{(t+1)} \leftarrow (1-\alpha_n) \hat{Q}(\boldsymbol{s}_i, \boldsymbol{d}_{i+1})^{(t)} + \alpha_n \max\{\hat{Q}(\boldsymbol{s}_i, \boldsymbol{d}_{i+1})^{(t)}, \hat{Q}(\boldsymbol{s}_{i+1}, \boldsymbol{d}_{i+2})^{(t+1)}\},
>      $$
>      where $\alpha_n$ is a weighting parameter that depends on the visit count $n(\boldsymbol{s}, \boldsymbol{d})$.
>    - **Visit Count**: The visit count $n(\boldsymbol{s}, \boldsymbol{d})$ is incremented for each direction.
>
> 6. **Scouting**:
>    - Feedback from each expansion is analyzed to extract general insights. These insights are stored in the global memory $\mathcal{I}$ and reused in future searches to guide direction generation. This enhances the algorithm’s exploitation capabilities.
>
> 7. **Scattering**:
>    - After each expansion, diverse new directions $\{d_1, d_2, \dots, d_k\}$ are generated for the newly created node using the LM. This encourages exploration of different regions of the solution space, reducing the likelihood of stagnation in local optima.
>
> 8. **Final Selection**:
>    - After $N$ iterations, the best solution $s^*$ across all trees in the forest is selected. The best solution is determined based on the highest reward (i.e., the **proportion of validation tests passed**).
>
> ---
>
> #### Key Features:
> - **Q-Value Initialization**: For every direction, **Q-values are initialized to 0** and updated dynamically during backpropagation.
> - **Reward Definition**: The reward is explicitly defined as the **proportion of validation tests passed**, providing a consistent and interpretable performance metric.
> - **Diversity**: The scattering technique ensures diversity in both initial seed solutions and improvement directions, allowing the algorithm to explore a wider solution space.
> - **Global Insights**: The scouting mechanism enables the sharing of effective strategies across branches, improving the efficiency of feedback exploitation.
>
> The integration of scattering, foresting, and scouting enables the SFS algorithm to balance exploration and exploitation effectively, making it a robust method for navigating complex search spaces.

---

> ### Author Response · Authors · 2024-11-23
> **Response 4**
>
> > Complicated method; Unclear contributions of each component: This paper includes many components in its method including (1) self-generated test cases, (2) itemized COT prompting, (3) UCB-based tree-expansion policy, (4) reusing COTs, (5) "foresting", and so on. It is unclear how each component contributes to the performance. I would suggest the authors perform ablation studies in a more common/standard setting, removing influences of components that are not part of this paper's contributions.
>
> Thank you for your thoughtful comment. We recognize the importance of isolating and understanding the contributions of each component in our method. Below, we provide a detailed breakdown of the ablation studies we have conducted, including additional experiments as per your suggestion.
>
> 1. **Ablation on Self-Generated Test Cases:**
>    The impact of self-generated validation tests is shown in **Table 8** (reproduced below). We also provide more detailed metrics in **Appendix I**. The results indicate that while using ground-truth tests improves performance, the improvement is relatively modest, underscoring the robustness of our method to noisy or self-generated tests.
>
>    | **Tests given**   | **pass@1** | **pass@any** | **BERT sim.** | **val. score** | **iters. (incl)** | **iters. (excl)** |
>    |--------------------|------------|--------------|---------------|----------------|-------------------|-------------------|
>    | No tests given     | 82.5%      | 89.0%        | 0.9945        | 0.813          | 1.68              | 5.06              |
>    | 3 tests given      | 87.2%      | 90.2%        | 0.9952        | 0.862          | 2.34              | 6.86              |
>    | All tests given    | 89.0%      | 90.2%        | 0.9949        | 0.864          | 2.04              | 5.88              |
>
> 2. **Ablation on Tree-Expansion Policies (PUCT vs. UCT):**
>    We conducted ablations on the tree-expansion policy used in MCTS to investigate its impact. As shown in **Appendix M** (Figure 22) and reproduced below, both UCT and PUCT achieve comparable performance under our method, although UCT performs slightly better in our settings.
>
>    | **Method**       | **pass@1** | **pass@any** | **BERT sim.** | **val. score** | **iters. (incl)** | **iters. (excl)** |
>    |-------------------|------------|--------------|---------------|----------------|-------------------|-------------------|
>    | Tree (PUCT)       | 73.8%      | 75.0%        | 0.9996        | 0.799          | 2.84              | 9.46              |
>    | Tree (UCT)        | 75.6%      | 76.9%        | 0.9998        | 0.827          | 2.61              | 8.87              |
>    | Ours (PUCT)       | 78.8%      | 86.3%        | 0.9943        | 0.788          | 2.13              | 5.88              |
>    | Ours (UCT)        | 82.5%      | 89.4%        | 0.9945        | 0.813          | 1.68              | 5.06              |
>
> 3. **Ablation on Key Components (Scattering, Foresting, and Scouting):**
>    We examined the contributions of the three main components introduced in our method: Scattering, Foresting, and Scouting. As shown in **Table 7** (reproduced below) and detailed in **Appendix H**, the ablations confirm that each component is necessary for optimal performance.
>
>    | **Ablation**     | **pass@1** | **pass@any** | **BERT sim.** | **val. score** | **iters. (incl)** | **iters. (excl)** |
>    |-------------------|------------|--------------|---------------|----------------|-------------------|-------------------|
>    | Everything        | 82.5%      | 89.0%        | 0.9945        | 0.813          | 1.68              | 5.06              |
>    | No *scatter*      | 75.6%      | 78.1%        | 0.9982        | 0.802          | 2.43              | 8.82              |
>    | No *forest*       | 79.4%      | 86.3%        | 0.9982        | 0.817          | 2.05              | 6.56              |
>    | No *scout*        | 81.9%      | 86.3%        | 0.9942        | 0.792          | 2.12              | 6.05              |
>
>    These results show that each component contributes significantly to performance improvements, highlighting the importance of Scattering for diversity, Foresting for exploration, and Scouting for leveraging feedback.
>
> 4. **Clarification on Itemized CoT Prompting and Reusing CoTs:**
>    We do not utilize itemized CoT prompting or reuse CoTs in our method. If any part of the paper suggests otherwise, we would greatly appreciate clarification from the reviewer so that we can revise and improve the presentation for better clarity.
>
> We hope these ablations and clarifications address your concerns about the contributions of each component. Thank you again for your valuable feedback, which has helped us improve the rigor and transparency of our work.

---

> ### Author Response · Authors · 2024-11-23
> **Response 5**
>
> > Are there clearer descriptions of the method, from the overall pipeline to each component such as the reward/value design?
>
> We added a more detailed description of the method in Appendix C. This includes pseudocode for how the algorithm works and a description of the pipeline for each component. For example, we explain in the description how the validation tests are used to provide a reward signal during the search process.
>
> > Are there results without the self-generated test cases which are neither this paper's contribution nor part of standard benchmarks?
>
>
> Yes, we also present results when ground truth tests are given (i.e. without self-generated test cases), as shown in table 8 and explained in more detail in appendex I. We see similarly strong performance of our method in this setting.
>
> > Are there results of more popular baselines than simple line-search?
>
> We compare to popular baselines such as LDB, LATS, Reflexion, and Best-of-N. LATS is the current state of the art baseline on the datasets we tested on, achieving the highest performance [14]. Best-of-N is the most popular approach to LLM code generation due to its ease of usage [5, 12]. We have also included results for the baselines that the reviewer suggested, as explained previously, including FunSearch, REx, and Self-repair.
>
> > Are there more detailed descriptions of the experimental settings, including how many LLM requests are used?
>
>  We have added these details to **Appendix C**, and have provided a summary of the pipeline previously.
>
>
> > Are there ablation studies in a cleaner or more standard setting?
>
> Yes, we conduct ablation studies on the model choice, components of our method, self-generated tests, and MCTS selection method as explained previously.
>
> ---
>
> [1] Romera-Paredes, Bernardino, et al. "Mathematical discoveries from program search with large language models." Nature 625.7995 (2024): 468-475.
>
> [2] Tang, Hao, et al. "Code Repair with LLMs gives an Exploration-Exploitation Tradeoff." arXiv preprint arXiv:2405.17503 (2024).
>
> [3] Olausson, Theo X., et al. "Is Self-Repair a Silver Bullet for Code Generation?." The Twelfth International Conference on Learning Representations. 2023.
>
> [4] Wang, Ruocheng, et al. "Hypothesis search: Inductive reasoning with language models." arXiv preprint arXiv:2309.05660 (2023).
>
> [5] Brown, Bradley, Jordan Juravsky, Ryan Ehrlich, Ronald Clark, Quoc V Le, Christopher Ré, and Azalia Mirhoseini. "Large language monkeys: Scaling inference compute with repeated sampling." *arXiv preprint* arXiv:2407.21787 (2024).
>
> [6] Chen, Bei, Fengji Zhang, Anh Nguyen, Daoguang Zan, Zeqi Lin, Jian-Guang Lou, and Weizhu Chen. "Codet: Code generation with generated tests." *arXiv preprint* arXiv:2207.10397 (2022).
>
> [7] Chen, Xinyun, Maxwell Lin, Nathanael Schärli, and Denny Zhou. "Teaching large language models to self-debug." *arXiv preprint* arXiv:2304.05128 (2023).
>
> [8] Islam, Md Ashraful, Mohammed Eunus Ali, and Md Rizwan Parvez. "Mapcoder: Multi-agent code generation for competitive problem solving." *arXiv preprint* arXiv:2405.11403 (2024).
>
> [9] Hong, Sirui, Xiawu Zheng, Jonathan Chen, Yuheng Cheng, Jinlin Wang, Ceyao Zhang, Zili Wang, Steven Ka Shing Yau, Zijuan Lin, Liyang Zhou, et al. "Metagpt: Meta programming for multi-agent collaborative framework." *arXiv preprint* arXiv:2308.00352 (2023).
>
> [10] Madaan, Aman, Niket Tandon, Prakhar Gupta, Skyler Hallinan, Luyu Gao, Sarah Wiegreffe, Uri Alon, Nouha Dziri, Shrimai Prabhumoye, Yiming Yang, et al. "Self-refine: Iterative refinement with self-feedback." *NeurIPS*, 36 (2023).
>
> [11] Shinn, Noah, Federico Cassano, Ashwin Gopinath, Karthik Narasimhan, and Shunyu Yao. "Reflexion: Language agents with verbal reinforcement learning." *NeurIPS*, 36 (2023).
>
> [12] Snell, Charlie, Jaehoon Lee, Kelvin Xu, and Aviral Kumar. "Scaling LLM test-time compute optimally can be more effective than scaling model parameters." *arXiv preprint* arXiv:2408.03314 (2024).
>
> [13] Zhong, Li, Zilong Wang, and Jingbo Shang. "LDB: A large language model debugger via verifying runtime execution step-by-step." *arXiv preprint* arXiv:2402.16906 (2024).
>
> [14] Zhou, Andy, Kai Yan, Michal Shlapentokh-Rothman, Haohan Wang, and Yu-Xiong Wang. "Language agent tree search unifies reasoning, acting, and planning in language models." *ICML*, 2024.

---

### Official Review · Reviewer_Jy4v · 2024-11-04

**Soundness:** 4
**Presentation:** 4
**Contribution:** 3
**Rating:** 6
**Confidence:** 4

**Summary:**

This paper focuses on scaling inference computing of language models for code generation. The paper proposes Scattered Forest Search (SFS), which involves the following steps: 1) scattering: choosing different search directions given a solution; 2) foresting: starting searching from different initial solutions; 3) scouting: sharing insights across search branches.

Empirically, SFS outperforms some well-accepted baseline algorithms: Best of N (BoN) and tree search, on HumanEval, MBPP, and other coding benchmarks.

**Strengths:**

This paper is well-written and easy to follow. The illustrations and examples are beneficial for understanding the paper.

The SFS algorithm is overall presented clearly. Although it’s similar to most search-based algorithms in the literature, it has a novel design and outperforms BoN and tree search empiriically. Ablation studies are done to understand the contributions of each component (Sec. E).

**Weaknesses:**

## Primary Concerns

The paper can be stronger with more thorough empirical evaluations and clearer justification on why it’s better than tree search. Specifically:

**Empirical results.** The number of solutions generated is small (up to 10). How does the algorithm perform with larger numbers of solutions (k)? In our words, do MCTS and BoN catch up with SFS with larger k’s?

I’m not sure if the authors have considered the strongest baselines. The authors seem to use the standard MCTS algorithm (Eq. (1)). Have the authors considered P-UCT, which considers the probability of direction $d$ prescribed by the language model? See Eq. (3) in [2].

**Comparison with tree search.** Can authors clarify why tree search algorithms produce similar results (caption of Fig. 3)? Tree search algorithms like MCTS should do explicit exploration, so they inherently perform the scattering step in SFS?

## Other Comments

**Presentation.**
It would be clearer to have an algorithm block to clarify how the three components work together. Specifically, how does the global memory work? Do you only keep the global memory for the last few steps (otherwise it may not fit in the prompt)?

**Related work.**
FunSearch [1] has a similar idea of exploring the search space which is not mentioned in the paper.

**Miscellaneous:**

Line 208: si, di -> s_i, d_i.

Line 241-242: The insights will then be store[d] in the global memory.

**References:**

[1] Romera-Paredes, Bernardino, et al. "Mathematical discoveries from program search with large language models." Nature 625.7995 (2024): 468-475.

[2] Zhang, Shun, et al. "Planning with large language models for code generation." arXiv preprint arXiv:2303.05510 (2023).

**Questions:**

See my questions in the weakness section above.

---

> ### Author Response · Authors · 2024-11-23
> **Response 1**
>
> > Empirical results. The number of solutions generated is small (up to 10). How does the algorithm perform with larger numbers of solutions (k)? In our words, do MCTS and BoN catch up with SFS with larger k’s?
>
> Thank you for this insightful question and suggestion. To address your question, we conducted additional experiments with a larger number of generated solutions, as detailed in Appendix D and shown in Figure 18. These experiments extend the number of solutions generated to 20, allowing us to analyze the scaling behavior of SFS compared to other methods.
>
> As depicted in Figure 18, SFS continues to demonstrate significant improvements in performance as the number of solutions increases, while the performance of other methods such as MCTS and BoN tends to plateau. Notably, even with increased inference scaling, neither MCTS nor BoN catches up to SFS in terms of performance. This result underscores the effectiveness of SFS in better exploring the code solution space, leveraging its ability to enhance diversity and incorporate execution feedback. SFS explores code solutions that other methods never consider, even with increased search budget.
>
> We appreciate the opportunity to further investigate this aspect and provide additional empirical evidence to strengthen the paper.
>
> > I’m not sure if the authors have considered the strongest baselines. The authors seem to use the standard MCTS algorithm (Eq. (1)). Have the authors considered P-UCT, which considers the probability of direction
>  prescribed by the language model? See Eq. (3) in [2].
>
>  We sincerely thank the reviewer for pointing out the importance of considering PUCT as a baseline. In response to this valuable suggestion, we have conducted additional experiments comparing the performance of UCT and PUCT within our framework. The results of these experiments, along with the associated discussion, have been added to the revised manuscript (see Appendix M).
>
> As detailed in Appendix M, both UCT and PUCT were evaluated on a representative benchmark, including HumanEval. PUCT, as described in Equation 12, introduces a policy prediction term derived from the underlying language model (LLM). This term leverages the token probabilities predicted by the LLM to guide exploration. However, our experiments reveal that PUCT and UCT perform comparably in our specific setting.
>
> This result is primarily due to the following limitations of PUCT in our use case:
>
> 1. **Misalignment of token probabilities with global solution quality**: The LLM's token-level predictions, while effective for next-token generation, are not inherently optimized for guiding global search strategies during MCTS.
> 2. **Sparse alignment between token probabilities and successful completions**: For many HumanEval problems, the cumulative token probabilities provide suboptimal guidance, limiting the utility of the policy term in PUCT.
> 3. **Robustness of UCT without high-quality priors**: UCT relies purely on empirical \(q\)-values and visit counts, allowing it to perform robustly even in the absence of reliable priors.
>
> Empirical results (see Table 20 and Figure 22) demonstrate that while PUCT can theoretically enhance exploration through policy priors, the practical gains in our setting are limited due to the aforementioned reasons. Our method, combined with UCT, achieves higher pass@1 and validation scores compared to PUCT, underscoring the robustness of UCT in scenarios where the quality of policy priors is not strong. Table 20 is reproduced below for convenience:
>
> | **Method**       | **pass@1** | **pass@any** | **BERT sim.** | **val. score** | **iters. (incl)** | **iters. (excl)** |
> |-------------------|------------|--------------|---------------|----------------|-------------------|-------------------|
> | Tree (PUCT)       | 73.8%      | 75.0%        | 0.9996        | 0.799          | 2.84              | 9.46              |
> | Tree (UCT)        | 75.6%      | 76.9%        | 0.9998        | 0.827          | 2.61              | 8.87              |
> | Ours (PUCT)       | 78.8%      | 86.3%        | 0.9943        | 0.788          | 2.13              | 5.88              |
> | Ours (UCT)        | 82.5%      | 89.4%        | 0.9945        | 0.813          | 1.68              | 5.06              |
>
>
>
> We appreciate the reviewer’s suggestion to explore PUCT, as it allowed us to strengthen our experimental framework and provide a more comprehensive analysis of selection policies in MCTS.

---

> ### Author Response · Authors · 2024-11-23
> **Response 2**
>
> > Comparison with tree search. Can authors clarify why tree search algorithms produce similar results (caption of Fig. 3)? Tree search algorithms like MCTS should do explicit exploration, so they inherently perform the scattering step in SFS?
>
> Thank you for pointing this out. We understand the need for clarification and appreciate the opportunity to elaborate. The issue lies not with the tree search algorithm itself but with how new solutions are proposed by the LLM. While MCTS encourages exploration by prioritizing less-visited nodes in the tree, the diversity of exploration is fundamentally constrained by the similarity of the proposed solutions. Below, we address the key points in detail:
>
> 1. **LLM’s Proposal Mechanism and Limited Diversity:**
>    In traditional MCTS, exploration benefits from diverse proposed actions. However, in our setting, where the action space consists of complete code solutions, the LLM often generates highly similar responses given the same input prompt. Even if MCTS prioritizes exploration of less-visited nodes, the lack of inherent diversity in these nodes undermines effective exploration. This phenomenon is evident in our experiments (Section 3.5), which show that solutions generated from the same prompt tend to be highly similar.
>
> 2. **Challenges with Large Action Spaces:**
>    In our scenario, the action space is the set of all possible code solutions within a specified length—a significantly larger and more complex space compared to traditional token-level actions. Traditional MCTS struggles with such large action spaces because effective exploration and exploitation require expanding many unvisited actions. Since it is computationally infeasible to exhaustively explore all possible actions, selective expansion becomes crucial. Without mechanisms to enhance the diversity of these selected actions, exploration remains suboptimal.
>
> 3. **Solution-Level MCTS vs. Token-Level MCTS:**
>    A significant distinction between our approach and prior works (e.g., [2]) is the level at which MCTS operates. Prior works apply MCTS at the **token generation level**, where the LLM’s log-probabilities can guide the selection of top $k$ tokens, inherently introducing variation and leading to diverse outcomes. In contrast, our method applies MCTS at the **solution level**, where each action corresponds to a complete code solution. Simply selecting the first $k$ responses generated by the LLM leads to highly similar solutions, failing to adequately explore the solution space.
>
> 4. **Role of Scattering in SFS:**
>    The Scattering technique in SFS addresses this limitation by explicitly diversifying the prompts used for solution generation. By encouraging the LLM to generate solutions along varied improvement directions, Scattering ensures that the resulting candidate solutions are more diverse, thereby enabling MCTS to explore the solution space more effectively. As demonstrated in Table 1 (reproduced below) and Figure 11, the absence of Scattering (seed_theme=None) results in highly similar solutions across all similarity metrics, leading to poorer performance. Introducing Scattering significantly reduces solution similarity, which directly improves exploration and subsequent performance.
>
> | **Seed theme** | **pass@1** | **pass@any** | **tf-idf sim.** | **BERT sim.** | **lev. sim.** | **seq. sim.** | **val. score** |
> |----------------|------------|--------------|------------------|---------------|---------------|---------------|----------------|
> | None           | 72.5       | 75.0         | 0.9013           | 0.9976        | 0.8971        | 0.9361        | 0.7786         |
> | Jabberwocky    | 74.4       | 81.9         | 0.7559           | 0.9944        | 0.7749        | 0.8444        | 0.7658         |
> | Style          | 79.4       | 88.1         | 0.6826           | 0.9929        | 0.7119        | 0.7504        | 0.7548         |
> | Role           | 81.9       | 87.5         | 0.7734           | 0.9957        | 0.7907        | 0.8323        | 0.7649         |
>
>
>  5. **Empirical Validation:**
>    We have provided experimental evidence in Section 3.5 to substantiate this explanation. In addition, we have included a detailed analysis in the appendix to further clarify these points for readers. We hope this enhanced explanation addresses your concern comprehensively.
>
> Once again, we thank the reviewer for raising this important question, which allowed us to better explain the core contributions of SFS in improving exploration and performance in large action spaces. Hopefully this helps clarify the question.

---

> ### Author Response · Authors · 2024-11-23
> **Response 3**
>
> > Presentation. It would be clearer to have an algorithm block to clarify how the three components work together. Specifically, how does the global memory work? Do you only keep the global memory for the last few steps (otherwise it may not fit in the prompt)?
>
> We thank the reviewer for this suggestion and have included a more detailed description of the SFS and how the components work together in Appendix C. Since the global memory is iteratively updated by the LLM as a whole, it will naturally contain the most relevant information (i.e, the most recent steps). We prompt the LLM to keep the insights concise to avoid context window issues.
>
> > Related work. FunSearch [1] has a similar idea of exploring the search space which is not mentioned in the paper.
>
> We thank the reviewer for suggesting this related work. We have included it in the paper, and have implemented FunSearch for our coding setting and tested it. We reproduce our results shown in Table 6 below:
> | **Method**      | **pass@1** | **pass@any** | **BERT sim.** | **val. score** | **iters. (incl)** | **iters. (excl)** |
> |-----------------|------------|--------------|---------------|----------------|-------------------|-------------------|
> | Line            | 68.1%      | 83.1%        | 0.9992        | 0.795          | 2.09              | 7.13              |
> | Tree (MCTS)     | 75.6%      | 76.9%        | 0.9998        | 0.827          | 2.38              | 8.09              |
> | Best of N       | 73.8%      | 75.6%        | 0.9983        | 0.774          | 2.59              | 9.00              |
> | FunSearch [2]   | 74.4%      | 75.6%        | 0.9994        | 0.815          | 3.04              | 10.36             |
> | **Ours (SFS)**  | **82.5%**  | **89.0%**    | **0.9945**    | **0.813**      | **1.67**          | **5.06**          |
>
> By integrating FunSearch into our comparisons, we provide a broader perspective on state-of-the-art search strategies and further validate the effectiveness of our approach. Thank you again for bringing this to our attention.

---

> ### Comment · Reviewer_Jy4v · 2024-11-26
>
> Thanks to the authors for the detailed response to my questions. Adding a comparison with PUCT and FunSearch does make the empirical evaluation more solid. Also I appreciate the thorough discussion on comparison with tree search algorithms.
>
> I'd like to keep my score the same.

---

> > ### Author Response · Authors · 2024-11-29
> > **Thanksgiving thank you note**
> >
> > Thank you for your thoughtful feedback and for recognizing the updates we made in response to your questions. We’re glad the added comparisons with PUCT and FunSearch, as well as the discussion on tree search algorithms, strengthened the empirical evaluation.
> >
> > We sincerely appreciate your detailed review and your support in maintaining your positive score.

---

### Official Review · Reviewer_DZCq · 2024-11-04

**Soundness:** 3
**Presentation:** 3
**Contribution:** 3
**Rating:** 8
**Confidence:** 4

**Summary:**

This paper frames code generation as a black box optimization problem within the code space, and employ optimization-inspired techniques to enhance exploration of the space. They introduce SCATTERED FOREST SEARCH to enhance solution diversity while searching for solutions. Their method improves performance by 4-8% on HumanEval+ and HumanEval

**Strengths:**

- A timely and appropriate direct of exploration for competitive programming tasks
- Extensive experiments and ablations on different benchmarks
- Their proposed search mechanism provides a good boost in performance, also in comparison to other search methods

**Weaknesses:**

- All experiments are only based on gpt-3.5. Showing the performance on at least some SoTA open source llm is helpful to understand the generality of usage of this method.

**Questions:**

What is the actual cost of the tree-search base inference per instance in comparison to other prior search methods, for the same budget
Would be interesting to see whether such tree-search techniques prove helpful in popular challenging benchmarks like SWEBench

---

> ### Author Response · Authors · 2024-11-23
> **Response**
>
> > All experiments are only based on gpt-3.5. Showing the performance on at least some SoTA open source llm is helpful to understand the generality of usage of this method.
>
> Thank you for your suggestion. We understand the importance of evaluating our method on multiple models to demonstrate its generality. In addition to experiments with GPT-3.5, GPT-4o, and GPT-4o-mini, we have added an open-source LLama-3.1-instruct model, per your suggestion. These results are included in the main manuscript (Figure 14).
>
> As shown in Figure 14, weaker models such as GPT-3.5 and LLama-3.1-instruct exhibit stronger scaling behaviors when using our method, suggesting that inference-time scaling compensates for limited training quality. By contrast, more capable models like GPT-4o display comparatively diminished scaling improvements, highlighting a trade-off between model pretraining quality and the benefits of inference-time scaling.
>
> These results demonstrate that our method is not only compatible with OpenAI's models but also extends effectively to SoTA open-source LLMs. This further underscores the generality and adaptability of our approach.
>
> We hope this clarifies the breadth of our evaluation and appreciate your feedback.
>
>
> > What is the actual cost of the tree-search base inference per instance in comparison to other prior search methods, for the same budget
>
> Great points! Per the reviewer's suggestion, we have included a scaling chart (Figure 12, right) that shows how our method scales with token usage. We see that our method scales better and performs better than the baselines given the same token budget.
>
> > Would be interesting to see whether such tree-search techniques prove helpful in popular challenging benchmarks like SWEBench
>
> Great idea! Definitely, we completely agree that applying tree-search techniques, particularly in challenging benchmarks like SWEBench, is a compelling avenue for exploration. This is indeed a work in progress, as we aim to extend our framework to domains that involve dynamic interactions with the environment.
>
> The SFS framework is particularly well-suited for such benchmarks due to its ability to incorporate execution feedback into the solution generation process. SWEBench, which emphasizes active interaction with the environment, aligns naturally with SFS's core strengths. By leveraging feedback from intermediate executions to guide the search, SFS can dynamically refine its exploration strategy, potentially unlocking significant performance gains in these settings.
>
> We believe this line of research holds great promise and could open new avenues for applying tree-search-based methods to domains requiring active and iterative feedback loops. Thank you again for raising this point, and we look forward to sharing further results as we continue to investigate this exciting direction.

---

### Author Response · Authors · 2024-11-23
**Summary of revisions**

We would like to thank all the reviewers for their insightful comments and for helping us improve our paper. Your feedback has been instrumental in refining the presentation, bolstering the experimental analysis, and providing a clearer explanation of our methodology. Below, we highlight the major revisions we’ve made in response to your feedback:

1. **Improved Clarity and Method Presentation**:
   - We have added detailed explanations of our methodology in **Appendix C**, including pseudocode and descriptions of how the core components—Scattering, Foresting, and Scouting—work together. This addresses concerns raised by reviewers (e.g., **hhUS** and **4VU2**) regarding the need for a clearer overall workflow and additional details on implementation.
   - A preliminary section on Monte Carlo Tree Search (MCTS) has been added in **Appendix C.1**, as suggested by **4VU2**, to help readers unfamiliar with MCTS understand its role in our framework.
   - A space-time complexity analysis of our algorithm has been added to **Appendix C.5**, as suggested by **4VU2**.

2. **Experimental Settings**:
   - As per **zwDf**'s feedback, we clarified our experimental setup in **Appendix B**, explaining the rationale for using self-generated test cases and how these align with real-world scenarios where ground-truth tests may not be accessible.
   - We also conducted experiments with ground-truth validation tests, as shown in **Table 8** and detailed in **Appendix I**. This provides a direct comparison of performance with and without self-generated test cases.

3. **Additional Baselines**:
   - In response to **zwDf**'s and **Jy4v**'s suggestions, we compared our method against additional baselines, including **FunSearch**, **REx**, and other state-of-the-art methods like **Self-repair** and **LDB**. These results are summarized in **Tables 5, 6, and 9**. We find that SFS consistently outperforms these baselines across multiple metrics, further demonstrating the robustness of our approach.

4. **Ablation Studies**:
   - We conducted extensive ablation studies to evaluate the contributions of our method's key components, including Scattering, Foresting, and Scouting.(e.g., **Table 7** and **Appendix H**). These studies show that each component contributes significantly to performance improvements.
   - Additionally, following **Jy4v**'s suggestion, we performed new ablations to compare **PUCT** and **UCT** tree-expansion policies, as detailed in **Appendix M**, highlighting why UCT performs better in our specific setting.
   - Per reviewer **DZCq**'s suggestion, we have included a comparison against open-source models such as LLama-3.1


5. **Scaling Experiments**:
   - We included a scaling chart (**Figure 12, right**) to analyze how our method performs under different token budgets, addressing **DZCq**'s request for a cost analysis of tree-search-based inference. These results demonstrate that SFS scales more effectively than baselines given the same token budget.

6. **Visualization of Local Optima Escape**:
   - To respond to **hhUS**'s request for a clearer demonstration of how our method avoids local optima, we added a new illustration (**Figure 19**) and theoretical insights in **Appendix L**. This visualization shows how text-based improvement directions enable transitions between clusters in the solution space, facilitating the escape from local optima.

7. **Improved Presentation**:
   - We have addressed **4VU2**'s comments on notation inconsistencies and improved the clarity of equations, including fixing typos in Eq. (1) and standardizing subscripts throughout the text.
   - Figures have been updated to PDF format for better readability, as suggested by **4VU2**.

8. **Tradeoff Discussion**:
   - Inspired by **hhUS**'s insightful comments, we added a discussion in the **Related Works** section and **Appendix F** on the tradeoff between training stronger models and optimizing inference-time performance. This contextualizes our work and highlights its complementary role to model training.

We believe these revisions comprehensively address the reviewers’ feedback and significantly improve the quality of the paper. Thank you again for your invaluable suggestions and for helping us make this work stronger and more impactful.

---

### Meta-Review · Area_Chair_KRQk · 2024-12-20

**Metareview:**

Summary:
This paper presents a novel search strategy employing LLM-based operators for code generation. It introduces an innovative prompting method to enhance the diversity of LLM outputs. Specifically, the approach first prompts the LLM to generate a list of chain-of-thoughts (CoTs) and subsequently uses each CoT to guide the generation of corresponding code. The itemized prompting results in more diverse CoTs, which in turn lead to greater diversity in the generated code. Additionally, the method caches CoTs for reuse across different code generation tasks. Another key contribution is the use of a forest of search trees to "encourage exploration" during the search process. The proposed approach is compared against simple baselines, demonstrating improved performance in a customized evaluation setting.

Strengths:
(1) The proposed method looks sound and novel. (2) The author provides a theoretical foundation for the approach. (3) The experiments are thorough, and the method demonstrates strong empirical performance.

Weakness:
This work does not touch much on the design of training dataset, training paradigm and training objective.

Decision:
While the above-mentioned weakness is a notable concern, I believe the novelty and contributions of this paper outweigh it. I recommend acceptance; however, I strongly encourage the authors to consider how their work could contribute to improvements in the training dataset, training paradigm, and training objectives in future iterations.

**Additional Comments On Reviewer Discussion:**

Four out of five reviewers gave positive ratings for this paper after the rebuttal. I believe the authors have addressed most of the concerns. The remaining reviewer, who assigned a score of 5, highlighted that the paper would have greater significance if it contributed more to the training dataset, training paradigm, and training objective rather than focusing primarily on the prompt. While I agree with this observation, I believe the merits of the paper outweigh its shortcomings. I encourage the authors to carefully address all reviewer comments when preparing the final version.

---

### Decision · Program_Chairs · 2025-01-22

Accept (Poster)